# PF△: A Benchmark Dataset for Power Flow under Load, Generation, and Topology Variations

**Ana K. Rivera**,* **Anvita Bhagavathula**\*, **Alvaro Carbonero**\*, **Priya L. Donti**
Department of Electrical Engineering & Computer Science
Laboratory for Information & Decision Systems
Massachusetts Institute of Technology
{akrivera, abhagava, alvarocg, donti}@mit.edu

## Abstract

Power flow (PF) calculations are the backbone of real-time grid operations, across workflows such as contingency analysis (where repeated PF evaluations assess grid security under outages) and topology optimization (which involves PF-based searches over combinatorially large action spaces). Running these calculations at operational timescales or across large evaluation spaces remains a major computational bottleneck. Additionally, growing uncertainty in power system operations from the integration of renewables and climate-induced extreme weather also calls for tools that can accurately and efficiently simulate a wide range of scenarios and operating conditions. Machine learning methods offer a potential speedup over traditional solvers, but their performance has not been systematically assessed on benchmarks that capture real-world variability. This paper introduces **PF△**, a benchmark dataset for power flow that captures diverse variations in load, generation, and topology. **PF△** contains 859,800 solved power flow instances spanning six different bus system sizes, capturing three types of contingency scenarios ($N$, $N$–1, and $N$–2), and including close-to-infeasible cases near steady-state voltage stability limits. We evaluate traditional solvers and GNN-based methods, highlighting key areas where existing approaches struggle, and identifying open problems for future research. Our dataset is available at `https://huggingface.co/datasets/pfdelta/pfdelta/tree/main` and our code with data generation scripts and model implementations is at `https://github.com/MOSSLab-MIT/pfdelta`.

## 1 Introduction

Solving the power flow (PF) problem – i.e., determining the voltages and powers of grid elements based on loads and generator outputs – is essential for operating and planning power systems [1]. For instance, performing contingency analysis (i.e., assessing the impacts of different equipment outages) requires solving thousands of PF problems across potential outage scenarios within operational timeframes (e.g., every 5 minutes) to support real-time decision making. Since solving AC power flow is too slow for such workflows, operators rely on faster linear approximations like DC power flow, which cause feasibility issues. Another important grid operations problem is topology optimization, which explores which discrete actions such as bus splitting or line switching can alleviate grid congestion; however, solving this problem involves searching a combinatorially large action space (e.g., a single substation on a 118-bus system can have 65,000 possible configurations [2]), and each search step relies on running PF. In addition, increasingly frequent climate-induced extreme weather events are also expected to cause more component outages, impacting grid resilience. To

---

*These authors contributed equally.

39th Conference on Neural Information Processing Systems (NeurIPS 2025) Track on Datasets and Benchmarks.

accommodate this increased uncertainty, grid operators must perform PF calculations at greater speed and scale to simulate outcomes across a diverse range of generation, load, and outage scenarios. While highly accurate, traditional PF solvers based on techniques such as Newton-Raphson (NR) incur high computational cost, making them impractical for such large-scale, real-time applications [3].

Machine learning (ML)-based approximators, particularly those based on graph neural networks, have emerged as an alternative to learn fast approximations to power flow, due to their fast runtime and inherent ability to represent the graph-structured nature of power grids [3–7]. However, despite its potential, today's research in this area faces two key limitations. The first is the absence of standardized benchmarking: different papers employ different data generation processes and evaluation approaches, making meaningful comparisons challenging. The second is that existing large-scale datasets only account for a subset of the variations real-world power grids are expected to face, such as uncertainty in generation and load profiles (due to time-varying renewables and changing energy demands) and changes in power grid topology (due to topology control actions and component outages from severe weather events). Developing models that are robust and reliable under these conditions and that scale to realistic grid sizes of $>1000$ nodes [8] is key to ensuring deployability. In light of these gaps, we make the following contributions:

1. We introduce **PF**$\Delta$, a benchmark dataset for evaluating ML approaches to power flow across variations in load distributions, generator profiles, grid sizes, and $N$–$1$/$N$–$2$ topological perturbations[2]. Our dataset also includes close-to-infeasible power flow cases near steady-state voltage stability limits that traditional solvers sometimes struggle to solve.

2. We propose a data generation scheme that combines the load sampling technique developed by [9], the topological perturbation scheme used by [10], and a novel approach for producing diverse generator profiles, in order to better capture diverse real-world variations.

3. We establish standardized tasks and metrics to assess performance across several regimes: in- and out-of-distribution generalization, data efficiency, and close-to-infeasible cases.

4. We evaluate the performance of a traditional power flow solver and three state-of-the-art graph neural network (GNN)-based approaches using the proposed tasks and metrics.

## 2   Related Work

**GNNs for power grids.** Several GNN-based power flow solvers have been proposed; however, different papers present unique training and evaluation regimes on custom datasets, making comparisons difficult. In addition, these custom datasets tend to only capture a subset of the variations experienced by real-world power grids, underscoring the need for realistic datasets and standardized training frameworks. For instance, *GraphNeuralSolver* [4], a self-supervised model that directly minimizes power balance equations in its loss function, is trained and evaluated over variations in generator/load profiles, line characteristics, and grid sizes, but not $N$–$1$ topological perturbations. *TypedGNN* [6], another self-supervised model, trains over variations in load, topology, and generator active and reactive power variations. However, both models are tested on a maximum grid size of 118 nodes. *PowerFlowNet* [7], a supervised model that combines bus-type mask embeddings with TAGConv message passing layers, trains over changes in load profile, generator setpoints, and grid size, and is evaluated on realistic grids, including the 6470-node French network [11]; however, it is not trained or evaluated on $N$–$1$ topological perturbations. GNNs for the closely related AC optimal power flow (ACOPF) problem have also been proposed.[3] One such example is *CANOS* [12], a supervised model that uses interaction networks and a combined L2 and constraint-violation loss. The model scales to large grid sizes (up to 10,000 nodes) and handles $N$–$1$ topological perturbations, but it is not trained on a realistically diverse set of load profiles or generator setpoints.

**Datasets for ACOPF.** Our work is inspired by recent efforts to create ML benchmarks for power grid problems. One such example is *PowerGraph* [13], a dataset designed for GNNs for power grids, supporting tasks such as power flow prediction and cascading failure analysis. While *PowerGraph* includes valuable real-world scenarios, such as ground-truth explanations for cascading failure events, it only supports a grid size of up to 118 buses and lacks explicit incorporation of topological

---

[2]An $N$–$k$ perturbation corresponds to the outage of $k$ grid components such as lines or generators.

[3]ACOPF optimally determines power generation setpoints given an objective function, AC power flow constraints, and device limits. AC power flow solves for grid state variables *given* power generation setpoints.

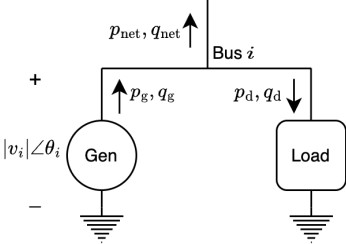

Figure 1: Illustration of a generator and load in a power system bus, labeled with each component's associated variables, adapted from [14].

Table 1: Bus types and their known (input) and unknown (output) variables in power flow analysis.

| Bus Type | Inputs | Outputs |
|---|---|---|
| Slack | $\lvert v \rvert, \theta$ | $p_{\text{net}}, q_{\text{net}}$ |
| Load (PQ) | $p_{\text{net}}, q_{\text{net}}$ | $\lvert v \rvert, \theta$ |
| Generator (PV) | $p_{\text{net}}, \lvert v \rvert$ | $\theta, q_{\text{net}}$ |

perturbations or variation in load and generator profiles. Other large-scale datasets for power grids focus on ACOPF. Notable examples include *OPFData* [10], a dataset capturing ACOPF solutions with load perturbations and $N$–1 topology perturbations, and *OPF-Learn* [9], a framework to create ACOPF datasets with diverse load profiles. While these datasets enable standardization and capture real-world variations, they are not well-suited out-of-the-box for benchmarking power flow. In particular, they lack diversity in generator setpoints, as ACOPF typically involves near-optimal setpoints (from a power cost perspective), whereas power flow must handle a wider range of setpoints.

## 3  Preliminaries: Power Flow

Given knowledge of power demand (load) and generator setpoints, the power flow problem aims to determine the voltages and power injections[4] for all nodes (buses) in a power system, as well as the power flows along all lines. Formally, we can represent the power network as a graph with nodes $\mathcal{B}$ and edges (lines) $\mathcal{L} \subseteq \{(i,j) \mid i,j \in \mathcal{B}, \ i \neq j\}$. Each node $i \in \mathcal{B}$ has four variables associated with it: $\lvert v_i \rvert$ (voltage magnitude), $\theta_i$ (voltage angle), $p_{\text{net},i}$ (net active power injection), and $q_{\text{net},i}$ (net reactive power injection). Figure 1 depicts how each of these variables are associated with a bus in a network. In the power flow setting, at each bus, two of these variables are known, while the remaining two are unknown. Depending on which quantities are known and unknown, nodes are classified as slack, load (PQ), or voltage-controlled generator (PV) buses (see Table 1).

Given the known bus quantities, power flow aims to determine all unknown quantities (i.e., all unknown bus variables and branch flows) to satisfy a specific set of non-linear equations. The nodal balance equations (1)–(2) reflect that the amount of power flowing into each bus must equal the amount of power flowing out; we define $\Delta p_i$ and $\Delta q_i$ as the active and reactive power balance mismatches, respectively, at each bus, and aim to set these quantities to 0. The line flow equations (3)–(4) compute the active and reactive power flows $p_{ij}$ and $q_{ij}$, respectively, along each line.

$$0 = \Delta p_i := p_{\text{net},i} - \lvert v_i \rvert \sum_{j=1}^{n} \lvert v_j \rvert \left( G_{ij} \cos \theta_{ij} + B_{ij} \sin \theta_{ij} \right) \quad \forall i \in \mathcal{B} \tag{1}$$

$$0 = \Delta q_i := q_{\text{net},i} - \lvert v_i \rvert \sum_{j=1}^{n} \lvert v_j \rvert \left( G_{ij} \sin \theta_{ij} - B_{ij} \cos \theta_{ij} \right) \quad \forall i \in \mathcal{B} \tag{2}$$

$$p_{ij} = \lvert v_i \rvert \lvert v_j \rvert (G_{ij} \cos \theta_{ij} + B_{ij} \sin \theta_{ij}) - G_{ij} \lvert v_i \rvert^2 \quad \forall (i,j) \in \mathcal{L} \tag{3}$$

$$q_{ij} = \lvert v_i \rvert \lvert v_j \rvert (G_{ij} \sin \theta_{ij} - B_{ij} \cos \theta_{ij}) + \lvert v_i \rvert^2 (B_{ij} - b_{s,ij}) \quad \forall (i,j) \in \mathcal{L}. \tag{4}$$

(Here, $\theta_{ij} := \theta_i - \theta_j$, and $G$ and $B$ are the real and imaginary components, respectively, of the network admittance matrix $Y$.) Traditional power flow solvers generally approach this problem in two stages: (a) calculate all unknown bus voltages by using an iterative solver such as Newton-Raphson to solve the nonlinear, implicit system of equations comprising (1)–(2) at PQ buses and (1) at PV buses, and (b) plug the given and calculated voltage quantities into the remaining equations to analytically

---

[4]Voltages and powers are complex-valued. Active (real) power flows in the direction of energy transfer and performs useful work; reactive (imaginary) power oscillates between source and load without doing useful work.

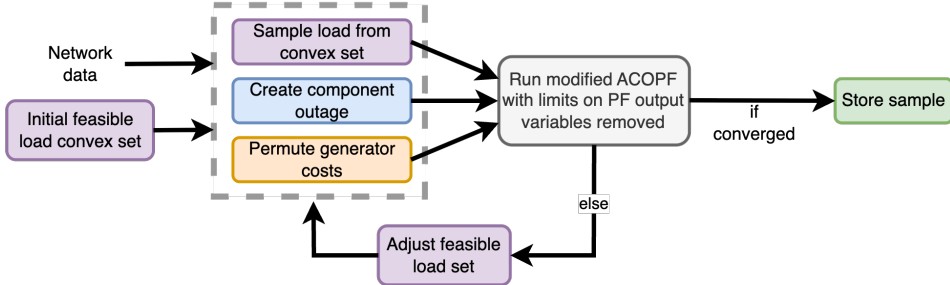

Figure 2: Data generation process for a single data sample within **PFΔ**.

calculate the rest of the network state – i.e., $p_{net}$ and $q_{net}$ at the slack bus, $q_{net}$ at all generator buses, and $p_{ij}$ and $q_{ij}$ at all lines. Some GNN-based approaches similarly adopt a two-stage approach, using a GNN to approximate step (a) and then plugging the resultant voltage predictions into the remaining equations; other approaches work end-to-end manner by directly predicting the full network state.

## 4 PFΔ: A Benchmark for Power Flow

We present **PFΔ**, a benchmark for ML-for-power-flow methods that captures perturbations in load profiles, generator profiles, and network topologies. **PFΔ** consists of a total of 859,800 solved AC power flow instances across the following bus system sizes: IEEE-14, IEEE-30, IEEE-57, IEEE-118, GOC-500, and GOC-2000 [15, 16]. We provide 159,600 samples per bus system (except for GOC-2000, for which we offer 61,800 samples[5]) that span three distinct feasibility regimes: (1) cases that are feasible, (2) cases that approach infeasibility, and (3) cases that are close-to-infeasible. We additionally present a set of standardized evaluation tasks to assess model performance across four regimes: in-distribution generalization to $N$-1 and $N$-2 topological perturbations, data efficiency, out-of-distribution generalization to different grid sizes, and performance on challenging close-to-infeasible cases.

### 4.1 Data Generation

Figure 2 illustrates our data generation workflow for a single data sample. The process starts by obtaining power network data for the grid size of interest, and then simultaneously introducing three types of perturbations relative to the "base case" reflected in the network data. The goal of applying these perturbations is to simulate data instances reflecting a wide range of grid conditions, to better facilitate the training and evaluation of learning methods.

- **Load perturbation:** We utilize the *OPF-Learn* load sampling method [9], which uniformly samples load profiles from a convex set that contains the ACOPF feasible space. As in [9], when an infeasible point is found, the convex set is reduced using infeasibility certificates. Unlike [9], which considers a more traditional ACOPF formulation, we utilize a custom ACOPF formulation tailored to the power flow problem to generate these points, where limits on output variables for power flow have been removed. A complete description of the formulation is provided in Appendix A.3. While a common alternative approach is to sample active power demand from a uniform distribution within $\pm$ 20% of the nominal base case (see, e.g., [10]), we find this does not result in sufficient load profile diversity in the generated power flow cases (see Appendix A.4).

- **Topological perturbation:** We induce $N$-1 and $N$-2 topological perturbations, i.e., simulate component outage events where up to two generators or lines (or one of both) are lost [17]. We build upon the approach proposed by [10], where each data sample is subject to one of the following equally probable events: (1) removal of up to two randomly chosen generators, (2) removal of up to two randomly chosen lines, (3) removal of a randomly chosen line and generator, or (4) no removal, maintaining the original topology.

- **Generator cost perturbation, for setpoint creation:** To create diverse active power and voltage setpoints on the generators of the system, we perturb the generator cost functions in the base-case

---

[5]We offer fewer samples for GOC-2000 due to the exponentially large computational cost of scaling to larger grid sizes, which we were not able to fully accommodate within our compute setup.

data. These cost functions are ultimately used to solve a modified version of ACOPF, resulting in different setpoints. Specifically, we randomly permute generator cost parameters (which specify how expensive it is for each generator to produce electricity) between and among generators.

Given a set of perturbed inputs, we aim to determine whether they correspond to a feasible power flow solution. In particular, to obtain a power flow solution, we pass the perturbed data sample through a custom formulation of ACOPF using PowerModels.jl [18] and Ipopt [19], a robust state-of-the-art solver. The custom ACOPF formulation removes limits on power flow output variables, namely reactive power generation, voltage magnitude at PQ buses, and branch flows, which makes the ACOPF constraints equivalent to the power flow equations. If ACOPF converges, the sample is accepted, as shown in Figure 2; in other words, we only accept inputs with feasible solutions into our dataset. In addition to saving the perturbed inputs, we also save the obtained solution in order to ensure our dataset is compatible with supervised learning methods; however, all benchmark evaluation metrics are unsupervised (see Section 4.4) to avoid biases associated with the particular solution obtained (see next paragraph). This data generation pipeline is fully generalizable to any network grid that is saved in a data format that can be parsed by PowerModels (such as PSS/E and MATPOWER formats).[6]

An important point to note is that power flow problems can have multiple solutions, and best practice is to select a physically meaningful, operational point. Rather than enumerating all solutions, which would require solving with different initializations and incur large computational cost, we follow standard practice in the optimal power flow and power flow literature by saving the solution the solver converges to. While we do not explicitly recover all power flow solutions, our data generation procedure captures diverse operating conditions, including high-voltage, small-angle-difference solutions (typical of normal operations) and lower-voltage, larger-angle-difference solutions (useful for studying system stress or instability). Appendix A.4 demonstrates that **PF$\Delta$** exhibits feature diversity reflective of these operating conditions in comparison to existing large-scale datasets *OPF-Learn* and *OPFData*.

## 4.2 Close-to-Infeasible Cases

We also generate **close-to-infeasible cases**, which are defined in terms of their proximity to steady-state voltage stability limits. These cases correspond to scenarios where the grid is at its loadability limit. At this point, a further increase in load cannot be accommodated by the grid. Traditional power flow solvers often struggle to converge on these cases, due to the problem being ill-conditioned. These samples are obtained by progressively increasing power injections and withdrawals in the network and making repeated power flow calculations. A close-to-infeasible case is then identified as being at the steady-state stability limit where the power flow Jacobian becomes singular; beyond this point, no power flow solution exists. Inspired by data augmentation strategies in ML, we also generate samples "approaching infeasiblity," representing points just before this boundary, for training. By computing such samples, we ensure that we include points in our dataset that Ipopt may not have converged to on its own. A detailed description of our procedure for computing close-to-infeasible cases can be found in Appendix A.1.

## 4.3 Dataset Summary

To the best of our knowledge, **PF$\Delta$** is the only power flow benchmark dataset that introduces simultaneous perturbations to loads, generator setpoints, and topologies, and that provides a data generation strategy for challenging cases that approach infeasibility; Appendix A.2 illustrates how our proposed dataset compares to prior benchmarks. A summary of the amount of data we generate for each topological perturbation and feasibility regime for each bus size is in Appendix A.5. In addition to providing the raw data produced by this data generation process (available on Hugging Face under a `CC-BY-4.0` license), we provide a flexible PyTorch InMemoryDataset class to enable efficient integration with learning methods. This class provides both a general bus-generator-load level formulation of the grid as well as one that contains bus types specific to the power flow problem, such as PV, PQ, and slack buses. An overview of what each data instance contains and how it is formatted using the InMemoryDataset is given in Appendix A.6.

---

[6]If needed, data generation functions in our codebase can also be adapted for OPF with custom objectives.

## 4.4 Standardized Evaluation Tasks and Metrics

**Tasks.** We provide a set of standardized evaluation tasks to evaluate in- and out-of-distribution performance, data efficiency, and performance on close-to-infeasible cases. In-distribution tasks assess whether models trained on a grid of a given size (with topological perturbations) can perform well on grids of the same size, mirroring scenarios when grid operators leverage a solver for a given grid [20]. Data efficiency tasks echo scenarios with limited historical data or distribution shifts due to evolving grid conditions [21]. Out-of-distribution tasks capture cases where models are deployed on unfamiliar grid sizes, such as when a grid expands [4], or as general-purpose solvers. Lastly, evaluation on close-to-infeasible cases assesses model robustness in solving power flow cases close to the steady-state voltage limits where traditional solvers sometimes struggle to converge [22]. A summary of all evaluation tasks is in Table 2.

**Metrics.** We propose key performance metrics for model evaluation. For a model to be deployable, it fundamentally needs to produce solutions that satisfy the power balance equations (1) and (2), i.e., ensure that the net injected power at each bus is equal to zero. The magnitude of the complex power mismatch $|\Delta S_i|$ at each bus, i.e., the deviation from satisfying power balance, can be computed by combining the active and reactive power mismatches $\Delta p_i$ and $\Delta q_i$, respectively, as calculated by using the power flow problem inputs and each method's outputs:

$$|\Delta S_i| = \sqrt{(\Delta p_i)^2 + (\Delta q_i)^2} \ \ \forall i \in \mathcal{B}. \tag{5}$$

Based on this value, we define evaluation metrics listed in Table 3, specifically the mean and maximum power mismatch across the dataset. We also evaluate the runtime of each model. Furthermore, we introduce interpretability metrics, such as statistics on nodal features, to examine whether the model learns plausible solutions (given, e.g., that power balance solutions may be non-unique, and some solutions may not be practically usable). The interpretability metrics are described in Appendix A.7. One motivation for using such unsupervised metrics is to address solver bias in the situation when multiple power flow solutions exist. Using an unsupervised metric allows us to evaluate models based on how well outputs satisfy the power flow equations, rather than introducing evaluation biases in favor of the specific solutions found by Ipopt during the data generation process.

## 5 Experiments and Evaluation

### 5.1 Tested Models

We use **PF**$\Delta$ to perform a comparative analysis between a traditional power flow solver and state-of-the-art GNN-based approaches. We choose GNNs because they are the most popular ML approach identified for power flow; they are a natural fit given that power flow is a node value prediction problem and that traditional solvers need to generalize across varying grid sizes. However, our benchmark is readily compatible with other types of methods.[7] For the traditional solver, we select PowerModels.jl's Newton-Raphson solver [18]; we run this model in a "flat-start" mode (i.e., all voltages initialized to 1). For the GNN-based approaches, we select *CANOS* [12], *PowerFlowNet* [7], and *GraphNeuralSolver* [4].

**CANOS** was originally proposed to solve ACOPF. It leverages an encoder-decoder structure combined with an interaction network, where features of each grid component (buses, generators, shunts, loads) are projected into distinct high-dimensional spaces prior to message passing [23]. Predictions are made on nodal features, the decoder enforces nodal box constraints via sigmoid bounding, and the outputs are used to analytically calculate branch flows (3)–(4). The model is trained on a combined L2 and constraint violation loss. As the original code is not publicly available, we re-implement *CANOS*, with our replicated results for ACOPF shown in Appendix A.8. We then adapt *CANOS* for power flow prediction, referring to our modified version as *CANOS-PF*. Key changes include customizing the encoder-decoder architecture for our dataset, removing sigmoid bounding, and modifying the constraint violation loss to focus solely on power and branch flow violations. Additionally, our predictions are used not only to derive branch flows analytically, but also the slack bus values.

---

[7]Node and edge data are stored as tensors, making it simple to access these attributes and feed them into any standard feedforward network.

| | Task | Topology | | Feasibility | | Bus Size | |
|---|---|---|---|---|---|---|---|
| **No.** | **Description** | **Train** | **Test** | **Train (# Samples)** | **Test** | **Train** | **Test** |
| **Task Group 1: In-Distribution Generalization** | | | | | | | |
| 1.1 | Unperturbed training topology | $N$ | $N$ $N-1$ $N-2$ | Feasible (54,000) | Feasible, Close-to-infeasible | $14, 30, 57, 118$ $500$, or $2000$ | Same as train |
| 1.2 | N-1 perturbed training topology | $N$ $N-1$ | $N$ $N-1$ $N-2$ | Feasible (54,000) | Feasible, Close-to-infeasible | $14, 30, 57, 118$ $500$, or $2000$ | Same as train |
| 1.3 | N-2 perturbed training topology | $N$ $N-1$ $N-2$ | $N$ $N-1$ $N-2$ | Feasible (54,000) | Feasible, Close-to-infeasible | $14, 30, 57, 118$ $500$, or $2000$ | Same as train |
| **Task Group 2: Data Efficiency** | | | | | | | |
| 2.1 | Low data efficiency | *Same experimental setup as Task 1.3* | | | | | |
| 2.2 | Medium data efficiency | $N$ $N-1$ $N-2$ | $N$ $N-1$ $N-2$ | Feasible (36,000) | Feasible, Close-to-infeasible | $14, 30, 57$ $118, 500$ or $2000$ | Same as train |
| 2.3 | High data efficiency | $N$ $N-1$ $N-2$ | $N$ $N-1$ $N-2$ | Feasible (18,000) | Feasible, Close-to-infeasible | $14, 30, 57$ $118, 500$, or $2000$ | Same as train |
| **Task Group 3: Out of Distribution Generalization to Different Grid Sizes** | | | | | | | |
| 3.1 | Fixed training grid size | $N$ $N-1$ $N-2$ | $N$ $N-1$ $N-2$ | Feasible (54,000) | Feasible, Close-to-infeasible | $14, 30, 57$ $118, 500$, or $2000$ | All of the others |
| 3.2 | Small grid size training group | $N$ $N-1$ $N-2$ | $N$ $N-1$ $N-2$ | Feasible (54,000) | Feasible, Close-to-infeasible | $14, 30,$ *and* $57$ | $118, 500,$ $2000$ |
| 3.3 | Large grid size training group | $N$ $N-1$ $N-2$ | $N$ $N-1$ $N-2$ | Feasible (54,000) | Feasible, Close-to-infeasible | $118, 500,$ *and* $2000$ | $14, 30, 57$ |
| **Task Group 4: Training on Challenging Power Flow Cases** | | | | | | | |
| 4.1 | Training with hard power flow cases | $N$ $N-1$ $N-2$ | $N$ $N-1$ $N-2$ | Feasible (48,600) + Close-to-infeas. (5,400) | Feasible, Close-to-infeasible | $14, 30, 57, 118$ $500$, or $2000$ | Same as train |
| 4.2 | Training with augmented hard power flow cases | $N$ $N-1$ $N-2$ | $N$ $N-1$ $N-2$ | Feasible (27,000) + Close-to-infeas. (5,400) + Approaching infeas. (21,600) | Feasible, Close-to-infeasible | $14, 30, 57, 118$ $500$, or $2000$ | Same as train |
| 4.3 | Training only with hard power flow cases | $N$ $N-1$ $N-2$ | $N$ $N-1$ $N-2$ | Close-to-infeas. (10,800) + Approaching infeas. (43,200) | Feasible, Close-to-infeasible | $14, 30, 57, 118$ $500$, or $2000$ | Same as train |

Table 2: Standardized evaluation tasks for **PF**$\Delta$ spanning in- and out-of-distribution generalization, data efficiency, and performance on close-to-infeasible cases. Training samples for each task are split evenly across all training topologies (e.g., Task 1.2 uses 54,000/2 samples for each of $N$ and $N-1$) and across all training bus sizes (e.g., Task 3.2 uses 54,000/3 samples per bus size). For each bus size, test sets contain 6,000 feasible and 600 close-to-infeasible samples across all tasks. For the GOC-2000 case, all sample counts are halved (see Appendix A.5).

**PowerFlowNet** (PFNet) is a supervised learning method that combines bus-typed embeddings of the input features and mask embeddings with TAGConv message passing layers. These layers are

| Power Balance Loss (Mean) | Power Balance Loss (Max) | Solver Runtime |
|---|---|---|
| $\text{Mean}_{\text{sample}}(\Delta S)$ $= \frac{1}{|\mathcal{B}|}\sum_{i\in\mathcal{B}}|\Delta S_i|$ | $\text{Max}_{\text{sample}}(\Delta S)$ $= \max(\{|\Delta S_i|, \forall i \in \mathcal{B}\})$ | $\dfrac{\sum_{b\in\text{test batches}}\text{runtime}(b)\times\text{size}(b)}{\sum_{b\in\text{test batches}}\text{size}(b)}$ |

Table 3: Metrics for evaluating model performance. Interpretability metrics are in Appendix A.7.

designed to leverage K-localized adaptive filters that extract multi-scale features across different receptive field sizes while preserving the graph's topological structure [24]. This enables better information propagation while reducing the risk of oversmoothing after several layers of message passing. The model trains on an L2 loss. We use the publicly available *PFNet* code for our results. Replicated results of the original paper are provided in Appendix A.9.

**GraphNeuralSolver** (GNS) is a self-supervised method that iteratively updates the GNN's nodal features during training by directly minimizing the power balance equations. It works in three stages. The first step calculates the global power consumption, which includes active power demand, power line losses, and shunt element consumptions. It also analytically determines a parameter $\lambda$ that adjusts every generator's active power output to ensure consumption and generation are equal. The second stage calculates the local power imbalance at each bus, which is used in the loss. Third, a neural network-based update is applied to the voltage magnitudes and angles based on the local power imbalance and messages aggregated from neighbors. As the original architecture modifies all generator active power outputs (i.e., the input values), it does not emulate a traditional power flow solver. Our version, *GNS-S*, modifies the architecture for our problem setup. Notably, in the first step, $\lambda$ is determined such that all PV generators maintain their pre-specified generation output, and only the slack bus output is modified. Our updated model formulation is provided in Appendix A.10.

## 5.2 Results

We select a representative subset of experiments from our standardized framework of tasks (see Table 2) to evaluate model performance across diverse operational conditions. Specifically, we focus on Tasks 1.1, 1.2, 1.3 (and 2.1), 3.1, 4.1, 4.2, and 4.3 which span a broad range of scenarios. Our primary analysis is conducted on the IEEE 118-bus system, which is typically the largest and most commonly used bus-system in the literature to train GNN-based power flow models. Before performing experiments, we conduct a grid-search based hyperparameter search strategy to tune each architecture's parameters (see details in Appendix A.11). To maximize architecture generalizability, hyperparameter tuning is performed using the training data from Task 1.3, which contains the most diverse set of topological variations within our dataset. Each model is trained three times with different random weight initializations and error bars are calculated as the standard deviations across these runs. All models were trained using either an NVIDIA V100 or NVIDIA RTX 8000 GPU. Key results are shown in Figure 3 and extended results are provided in Appendix A.12.

### 5.2.1 Task Group 1: In-Distribution Generalization

First, we examine how models generalize to $N$-1 and $N$-2 topological perturbations based on the diversity of their training topologies (Tasks 1.1–1.3). In general, we see that models trained only on the base ($N$) topology generally perform poorly on $N$-1 and $N$-2 cases. Even *CANOS-PF*, while achieving the best average performance across this task group, performs the worst on $N$-1 and $N$-2 cases when trained solely on $N$. This is likely due to the model's use of analytical formulas to derive branch flow and slack values from nodal predictions, an approach that amplifies errors when initial predictions are inaccurate. A key insight our results provide is that training on $N$-1 perturbations is improves generalization to both $N$-1 and $N$-2 cases across all models. We also note other interesting behavior such as *GNS-S*'s high error variance, possibly due to its constrained architecture that is sensitive to structural perturbations [4], and the fact that models with supervised components and message passing (especially interaction networks) exhibit robust performance, indicating that perhaps deeper and more expressive architectures are better equipped to learn representations under topological variation. Still, a key limitation remains: all models fail to achieve PBLs comparable to the Newton-Raphson (NR) solver (see Figure 4), even when they include physics-informed components, highlighting an open challenge in the field for architecture development.

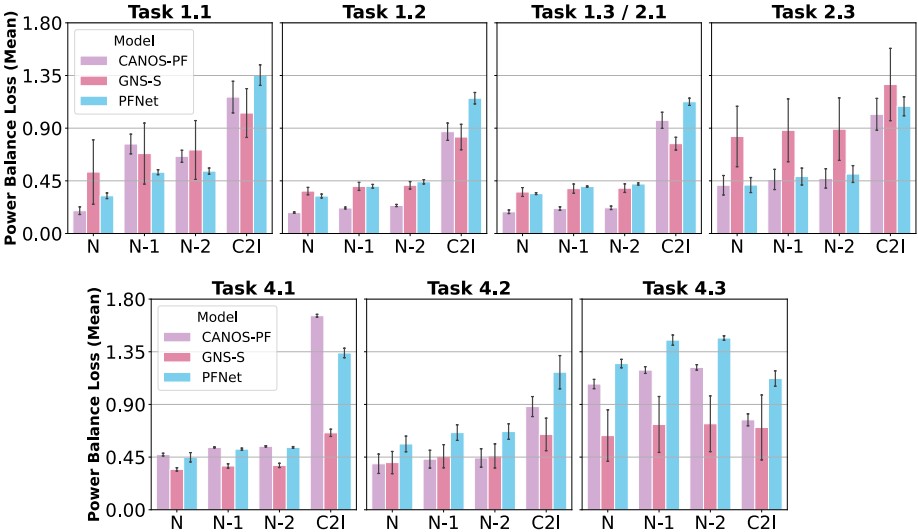

Figure 3: Experimental results for all selected tasks.

### 5.2.2 Task Group 2: Data Efficiency

Most models exhibit consistent performance across data efficiency tasks; performance patterns do not drastically change between the high-data and low-data regimes. An exception is *GNS-S*, which, in the low-data regime, performs comparably to when it is only trained on data without any topological perturbations. This result further suggests that topological diversity in the training data may play a more crucial role in driving model performance and improving generalization, compared to the amount of training data. However, as discussed earlier, the objective to develop models that can match the performance of NR solvers, even in these data-limited regimes, remains ground for future work in the field. This is especially a priority for real-time inference applications: models must be fast and adaptable to distribution shifts, making data-efficient, generalizable solvers essential for practical deployment.

### 5.2.3 Task Group 3: Out-of-Distribution Generalization and Runtime Analysis

Results for Task Group 3 are shown in Figure 4. We train on a 118-bus system and report test results on the feasible cases of a 57-bus system and a 500-bus system. We also include runtimes and comparisons to the NR solver. A note on the *PFNet* results is that the model employs normalization; we apply the same normalization parameters used for the training set to the other bus sizes. Overall, GNN-based models outperform NR in terms of runtime (NR runtime increases with the bus system size)[8]. The fastest model achieves approximately a $5\times$ speedup. This is in part because GNN-based models benefit from the hardware they can leverage, contributing to their speed advantage. However, even the best models do not achieve the $10^{-6}$ precision levels typical of NR. This experiment also highlights the challenge of generalizing to grids of a different size than the training grid; all three models struggle to do this. Achieving high accuracy on larger grids without relying on overly deep architectures (which suffer from oversmoothing) or designing models that can generalize across sizes, either through unique training strategies or architectural innovation, remain major open problems in this area.

### 5.2.4 Task Group 4: Performance on Close-to-Infeasible (C2I) Cases

Lastly, we examine model performance on close-to-infeasible (C2I) samples and the effect of including such cases in the training data. While *CANOS-PF* achieves the lowest average error overall in Tasks 4.1 and 4.2, *GNS-S* obtains the lowest C2I-specific losses across all of Task 4. This suggests that C2I samples are less out-of-distribution for *GNS-S* than for other models. In contrast, *PFNet*

---

[8]We only included runtimes for the samples that NR converged for; the samples that were not able to converge took longer, on average, to reach that point.

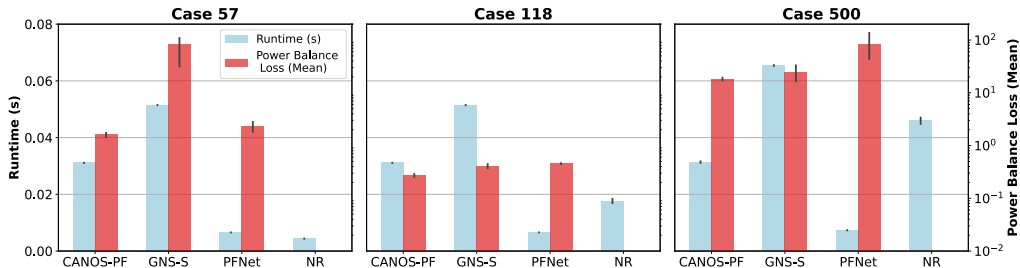

Figure 4: Results for Task 3.1 showcasing the Power Balance Loss (PBL) on a combined feasible and close-to-infeasible test set for three bus sizes. Runtimes and PBL of Newton-Raphson (NR) are reported only for the samples of the test set that converged from a flat start. Convergence rates are 95.2%, 65.7%, and 43.4% for bus sizes 57, 118, and 500, respectively, and PBL values for NR were approximately on the order of magnitude of $10^{-6}$. Runtime and PBL values for bus size 2000 are not included as NR failed to converge from a flat start for this system size. NR runtimes were calculated on an Intel Xeon Gold 6140 CPU and GNN runtimes were calculated on an NVIDIA RTX 8000 GPU.

consistently performs worst on these cases, highlighting the difficulty of handling near-infeasibility through purely supervised learning.

Interestingly, *GNS-S* maintains strong performance on feasible cases even when trained only on approaching infeasible and close-to-infeasible samples (Task 4.3), whereas *CANOS-PF* performance degrades slightly and *PFNet* performance reduces drastically in this regime. This could be attributed to the physics-informed components in *CANOS-PF* and *GNS-S* which promote prediction of solutions that satisfy power balance equations even under extreme operating conditions. Overall, performance on C2I samples remains limited when such points are excluded from training. Task 4.2 improves C2I performance but at the expense of feasible-case accuracy, while Task 4.3 highlights the opposite trend, except for *GNS-S*. These results underscore a key open challenge: designing models that reliably handle both normal and extreme operating conditions, potentially without needing C2I samples in their training sets to do so.

In addition to experiments on tasks, we also investigate the failure modes of the models. Key insights we take away from our analysis are that *CANOS-PF* is more consistent and stable across scenarios, while *PFNet* and *GNS-S* exhibit higher variability and occasional qualitatively different failure modes. A more detailed overview of this is provided in Appendix A.13.

## 6   Future Directions

**PF**$\Delta$ offers a standardized framework for evaluating ML-based power flow methods, serving as a valuable tool for grid operators and planners. This benchmark highlights several open challenges and research directions. A primary priority is the development of fast, scalable, and accurate solvers that support real-time decision-making under diverse grid conditions. Key goals include achieving feasibility on par with Newton-Raphson (NR) methods and generalizing to large-scale grids with $>$ 1000 buses. Another interesting direction is building models capable of reliably identifying infeasible power flow cases. While our dataset includes samples near the loadability limit, there is currently no benchmark to assess a model's ability to detect when a solution does not exist. Since NR solvers do not indicate a reason for non-convergence, ML models could fill this gap by identifying the reason for such cases. Other challenges also include designing architectures robust to higher-order topological perturbations, i.e., $N - k$ contingencies for $k > 2$. Given the combinatorial growth in possible outages as $k$ increases, ensuring robust model performance across a meaningful subset of these scenarios remains an important next step. On the data generation and benchmarking front, a key priority is evaluating models on out-of-distribution load profiles. Although [9]'s method samples from the full feasible load space, accurately representing this space in large networks is computationally infeasible. Another significant challenge is to develop scalable data generation methods that achieve high feature diversity for networks with thousands of nodes. Addressing this scalability issue is essential, since robust performance on large-scale networks is critical for real-world model deployment.

## Acknowledgments and Disclosure of Funding

We would like to thank Lara Booth for helping us create the name of this benchmark. This work was supported by the U.S. Department of Energy, Office of Science, Office of Advanced Scientific Computing Research, Department of Energy Computational Science Graduate Fellowship under Award Number DE-SC0025528; the Siddhartha Banerjee First-Year Fellowship at MIT; the Grass Instruments Fellowship at MIT; and the U.S. National Science Foundation (award #2325956).

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

# A    Technical Appendices and Supplementary Material

## A.1    Close-to-Infeasible Case Generation

Close-to-infeasible cases correspond to operating conditions near the steady-state voltage stability limit. Beyond this limit, no power flow solution exists, and the system is susceptible to voltage collapse, in which the whole system goes down. At this critical point, the power flow Jacobian becomes singular; solvers such as Newton-Raphson are prone to divergence [1, 25].

A technique commonly used to find the steady-state stability limit is *continuation power flow*, which traces a path of solutions to the power flow equations employing a predictor-corrector scheme. Specifically, the basic power flow equations $g(x)$, shown in Equation A.1 are reparameterized with a continuation parameter $\lambda \in \mathbb{R}_{>0}$, resulting in the system in Equation A.2, where $x = \begin{bmatrix} \theta^T & |v|^T \end{bmatrix}^T$. The notation used here is consistent with that used in [26], where $P(x)$ and $Q(x)$ corresponds to the second term in Equations 1 and 2, respectively, and $P^{\text{inj}}$ and $Q^{\text{inj}}$ are the net active and reactive power injections, respectively.

$$g(x) = \begin{bmatrix} P(x) - P^{\text{inj}} \\ Q(x) - Q^{\text{inj}} \end{bmatrix} = 0, \tag{A.1}$$

$$f(x, \lambda) = g(x) - \lambda \begin{bmatrix} P^{\text{inj}}_{\text{target}} - P^{\text{inj}}_{\text{base}} \\ Q^{\text{inj}}_{\text{target}} - Q^{\text{inj}}_{\text{base}} \end{bmatrix} = 0. \tag{A.2}$$

For a current solution $(x^j, \lambda^j)$, the predictor estimates the next point $(\hat{x}^j, \hat{\lambda}^j)$, typically by taking a step along the tangent direction of the solution trajectory. The corrector then uses this as a warm-start point for Newton's method in order to find the next solution $(x^{j+1}, \lambda^{j+1})$. If the corrector fails, the prediction has surpassed the power flow solvability boundary. The continuation path bends back at this point, forming a characteristic 'nose' shape (see Figure A.1).

We use MATPOWER's [26] continuation power flow implementation to trace the solution path. Specifically, we define $(P^{\text{inj}}_{\text{base}}, Q^{\text{inj}}_{\text{base}})$ as the setpoints of the load and generator of a randomly selected sample. The target injections are scaled as $(P^{\text{inj}}_{\text{target}}, Q^{\text{inj}}_{\text{target}}) = (2.5 \times P^{\text{inj}}_{\text{base}}, 2.5 \times Q^{\text{inj}}_{\text{base}})$. For each base case, a close-to-infeasible case is saved when the continuation method reaches the steady-state stability limit, identified by MATPOWER's "NOSE" event. To enrich the training set, we also include samples "approaching infeasibility" which correspond to the last four samples before the nose point was triggered. Only samples for which the NOSE event occurred or the continuation power flow converged successfully are retained. Figure A.1 illustrates an example path of power flow solutions traced for the IEEE-118 case, where the base case ($\lambda = 0$) corresponds to a sample drawn from the training set. Figure A.2 shows the condition numbers of the power flow Jacobian for the last 10 points of the curve. The last point corresponds to the steady-stability limit, which shows a much higher condition number, thus exhibiting the singularity of the Jacobian at this operating condition.

## A.2    Dataset Characteristics Comparison

Table A.1 compares **PF$\Delta$** with existing power flow datasets, including large-scale benchmarks and datasets designed for training specific GNN architectures. The comparison focuses on how these datasets meet key real-world deployability criteria, such as inclusion of load profile distributions, generator setpoint variations, varying grid sizes, N-1 and N-2 topological perturbations, close-to-infeasible cases, and sufficiently complex and realistic network sizes (>1000 buses).

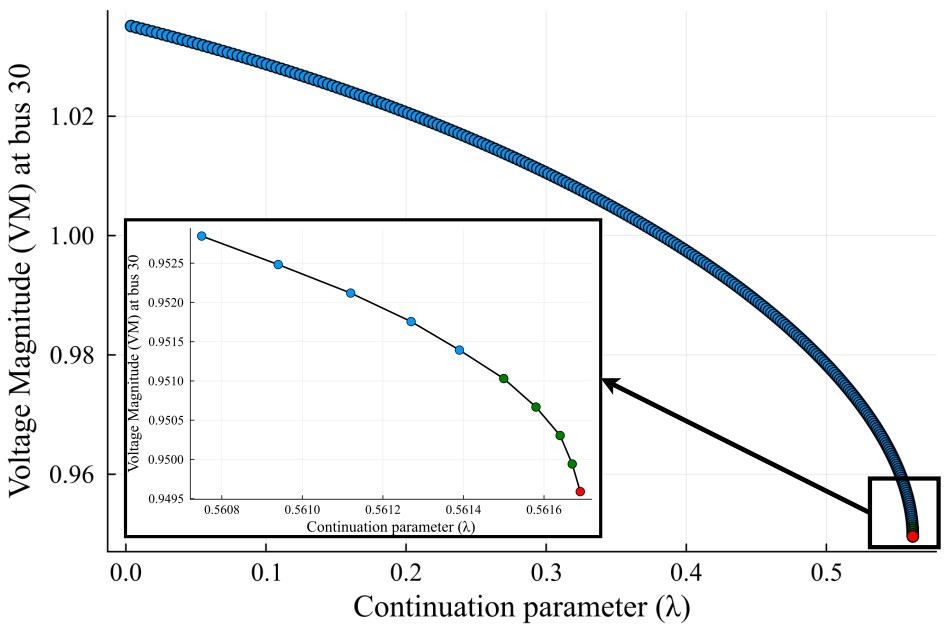

Figure A.1: Voltage magnitude at a load bus as a function of the continuation parameter $\lambda$. The point in red corresponds to the sample saved as close-to-infeasible, while the green points are samples labeled as "approaching infeasibility" and used to augment the training data.

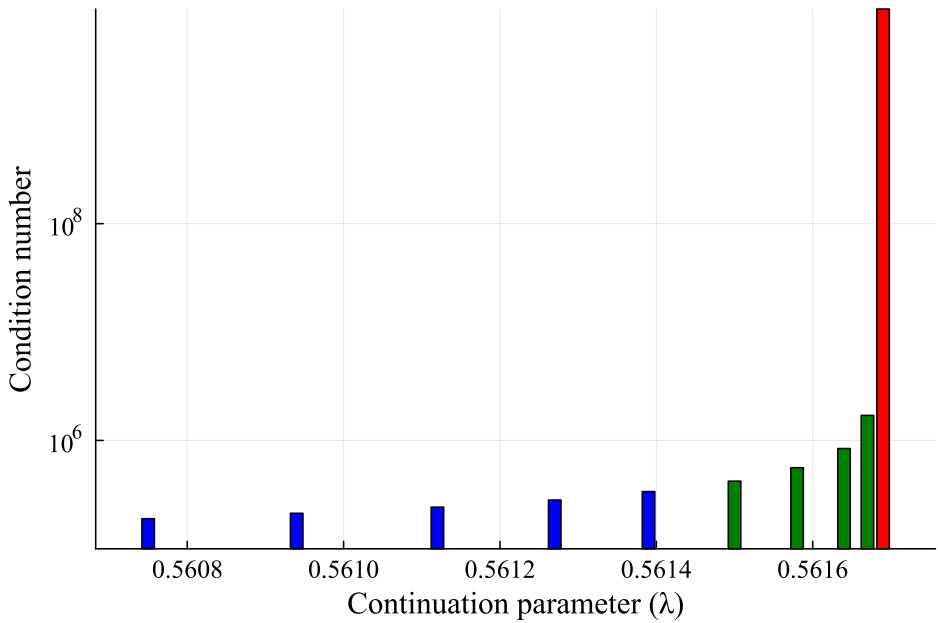

Figure A.2: Condition number of the power flow Jacobian as a function of the continuation parameter $\lambda$ (y-axis is on a log scale). We see increasing numerical sensitivity as the voltage stability limit is approached, with a sharp rise signaling the onset of ill-conditioning near the solvability boundary.

| Dataset | Load Profiles | Generator Profiles | Grid Sizes | N-1 | N-2 | Close to Infeasible | >1000 Buses |
|---|---|---|---|---|---|---|---|
| OPFData | ✓ | ✗ | ✓ | ✓ | ✗ | ✗ | ✓ |
| OPFLearn | ✓ | ✗ | ✓ | ✗ | ✗ | ✗ | ✓ |
| PowerGraph | ✗ | ✗ | ✓ | ✗ | ✗ | ✗ | ✗ |
| GraphNeuralSolver | ✓ | ✓ | ✓ | ✗ | ✗ | ✗ | ✗ |
| PowerFlowNet | ✓ | ✓ | ✓ | ✗ | ✗ | ✗ | ✓ |
| CANOS | ✓ | ✗ | ✓ | ✓ | ✗ | ✗ | ✓ |
| **PF**△ | ✓ | ✓ | ✓ | ✓ | ✓ | ✓ | ✓ |

Table A.1: Comparative assessment of benchmark datasets (OPFData, OPFLearn), custom datasets used to train GNN models, and **PF**△ in considering key criteria for real-world deployment.

### A.3 Custom ACOPF Formulation for Data Generation

For our data generation, we consider a custom formulation of ACOPF tailored to the power flow problem setting described in Equation (A.3). We adopt notation consistent with [16] where applicable. Let $\mathcal{B}$ denote the overall set of all buses, $\mathcal{D} \subseteq \mathcal{B}$ the set of PQ buses, $\mathcal{G} \subseteq \mathcal{B}$ the set of PV buses, and $\mathcal{R} \subseteq \mathcal{B}$ the set of slack (or reference) buses. The set $\mathcal{L}$ denotes the directed set of branches, where each $(i, j) \in \mathcal{L}$ indicates a branch with "from" bus $i$ and "to" bus $j$. The reverse orientation of the branches is captured by $\mathcal{L}^{\mathrm{R}}$. For a given bus $i$, let $\mathcal{L}_i$ denote the subset of edges incident to that bus.

$$\min_{p_g, q_g, |v|, \theta, p_{ij}, q_{ij}} \quad p_g^\top A p_g + b^\top p_g \tag{A.3a}$$

$$\text{s.t.} \quad p_{g_i}^{\min} \leq p_{g_i} \leq p_{g_i}^{\max} \qquad \forall i \in \mathcal{G} \setminus \mathcal{R} \tag{A.3b}$$

$$p_{g_i} \geq 0 \qquad \forall i \in \mathcal{R} \tag{A.3c}$$

$$|v_i|^{\min} \leq |v_i| \leq |v_i|^{\max} \qquad \forall i \in \mathcal{B} \setminus \mathcal{D} \tag{A.3d}$$

$$|v_i| \geq 0 \qquad \forall i \in \mathcal{B} \tag{A.3e}$$

$$\theta_i = 0 \qquad \forall i \in \mathcal{R} \tag{A.3f}$$

$$\theta_i - \theta_j \in [-\theta_{ij}^{\max}, \theta_{ij}^{\max}] \qquad \forall (i,j) \in \mathcal{L} \tag{A.3g}$$

$$p_{g_i} - p_{d_i} - g_s |v_i|^2 = \Re \left\{ \sum_{(i,j) \in \mathcal{L} \cup \mathcal{L}^{\mathrm{R}}} S_{ij} \right\} \qquad \forall i \in \mathcal{B} \tag{A.3h}$$

$$q_{g_i} - q_{d_i} + b_s |v_i|^2 = \Im \left\{ \sum_{(i,j) \in \mathcal{L} \cup \mathcal{L}^{\mathrm{R}}} S_{ij} \right\} \qquad \forall i \in \mathcal{B} \tag{A.3i}$$

$$S_{ij} = \left( Y_{ij}^* - i \frac{b_{ij}^c}{2} \right) \frac{|V_i|^2}{|T_{ij}|^2} - Y_{ij}^* \frac{V_i V_j^*}{T_{ij}} \qquad \forall (i,j) \in \mathcal{L} \tag{A.3j}$$

$$S_{ji} = \left( Y_{ij}^* - i \frac{b_{ij}^c}{2} \right) |V_j|^2 - Y_{ij}^* \frac{V_j V_i^*}{T_{ij}^*} \qquad \forall (i,j) \in \mathcal{L} \tag{A.3k}$$

$$\text{where } V_i = |v_i| e^{j\theta_i}, \quad Y_{ij} = \frac{1}{r_{ij} + i x_{ij}}, \quad T_{ij} \text{ is the complex tap ratio}$$

We consider a generation cost minimization objective. Equation (A.3b) enforces active power generation limits at PV buses only (i.e., only for power flow input-related quantities), while active power generation at slack buses and reactive power generation at both PV and slack buses (i.e., power flow output-related quantities) are left unconstrained. Equation (A.3d) bounds the voltage magnitude at PV and slack buses, while for PQ buses we only require them to be nonnegative. The rest of the constraints are as standard for ACOPF. The slack bus angle is fixed to zero in Equation (A.3f). Equation (A.3g) corresponds to voltage angle difference limits, Equations (A.3h) and (A.3i) enforce active and reactive power balance at each bus, and (A.3j) and (A.3k) ensure the branch flows follow Ohm's Law.

## A.4 Feature Diversity Comparison Across Benchmark Datasets

Figures A.3 - A.8 present violin plots comparing 10,000 samples from four datasets (PFΔ, Power-Graph, OPFData, and OPFLearn) for the IEEE 118-bus system. These plots illustrate the distribution of nodal variables across datasets, highlighting the diversity introduced by our data generation process. Specifically, we compare the spread of real power demand ($p_d$), reactive power demand ($q_d$), active power generation ($p_g$), reactive power generation ($q_g$), voltage magnitude ($|v|$), and voltage angle ($\theta$) across seven randomly selected nodes in the system.

Overall, our dataset exhibits comparable or greater variability in these quantities compared to existing large-scale benchmarks. While comparisons with the OPF datasets like OPFData and OPFLearn dataset are not entirely direct given its more constrained ACOPF formulation (which inherently limits variability in power generation values), our dataset maintains a broader distribution in several dimensions. In contrast, when compared to a power flow dataset like PowerGraph, PFΔ exhibits more variability in all the compared variables. Notably, for active power demand ($p_d$), the distribution in PFΔ closely matches that of OPFLearn, suggesting similar diversity in load sampling across the two datasets.

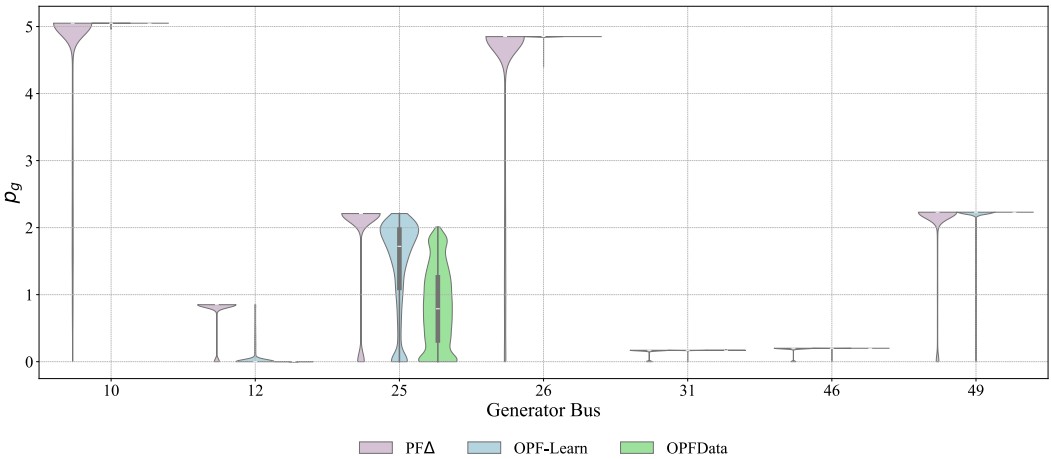

Figure A.3: Violin plots illustrating spread of $p_g$ values in 7 randomly selected generation buses. This graphic compares feature diversity of $p_g$ values sampled from $PF\Delta$ to other large-scale benchmark datasets. Note: PowerGraph has not been included in this plot, as this dataset does not report component-level active power generation values.

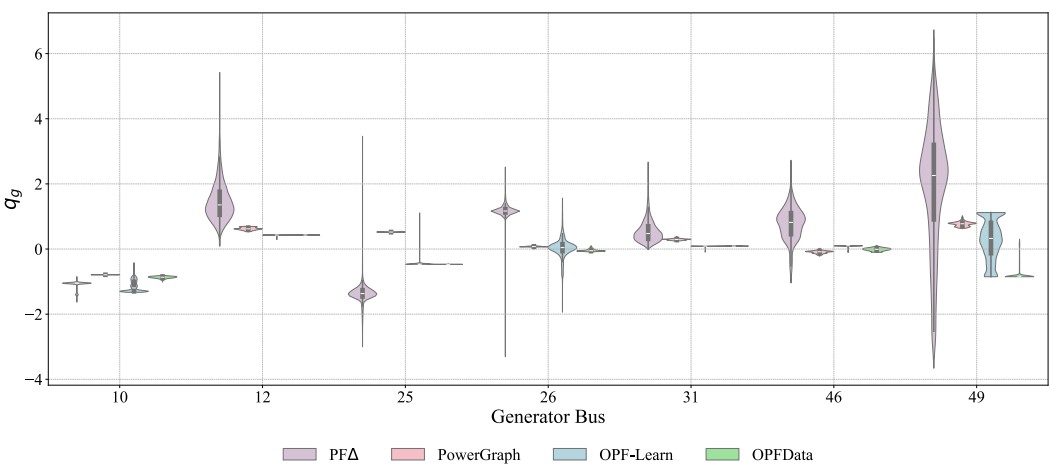

Figure A.4: Violin plots illustrating spread of $q_g$ values in 7 randomly selected generation buses. This graphic compares feature diversity of $q_g$ values sampled from $PF\Delta$ to other large-scale benchmark datasets.

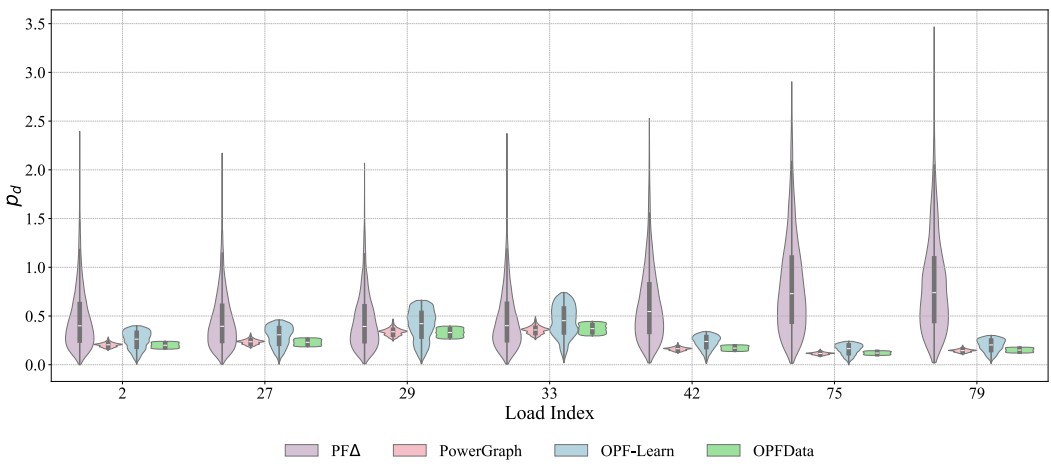

Figure A.5: Violin plots illustrating spread of $p_d$ values in 7 randomly selected loads at PQ buses. This graphic compares feature diversity of $p_d$ values sampled from $PF\Delta$ to other large-scale benchmark datasets.

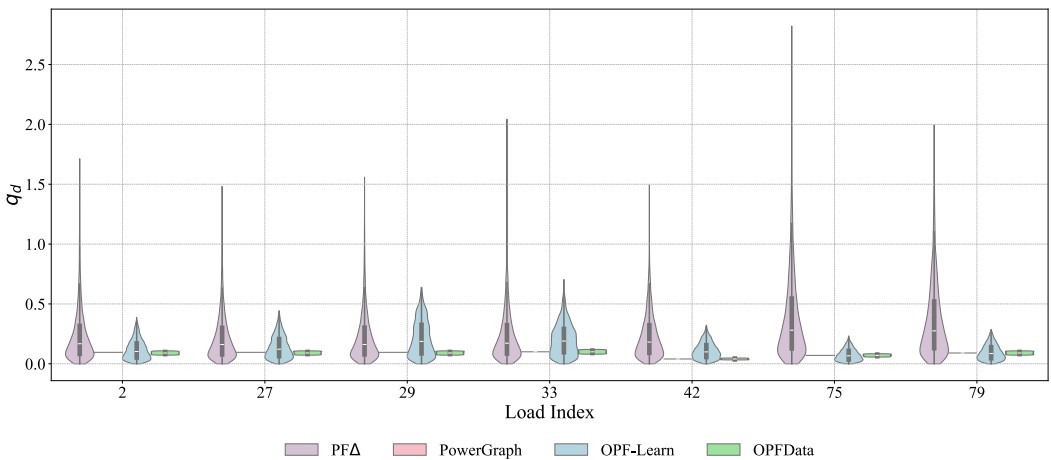

Figure A.6: Violin plots illustrating spread of $p_d$ values in 7 randomly selected loads at PQ buses. This graphic compares feature diversity of $q_d$ values sampled from $PF\Delta$ to other large-scale benchmark datasets.

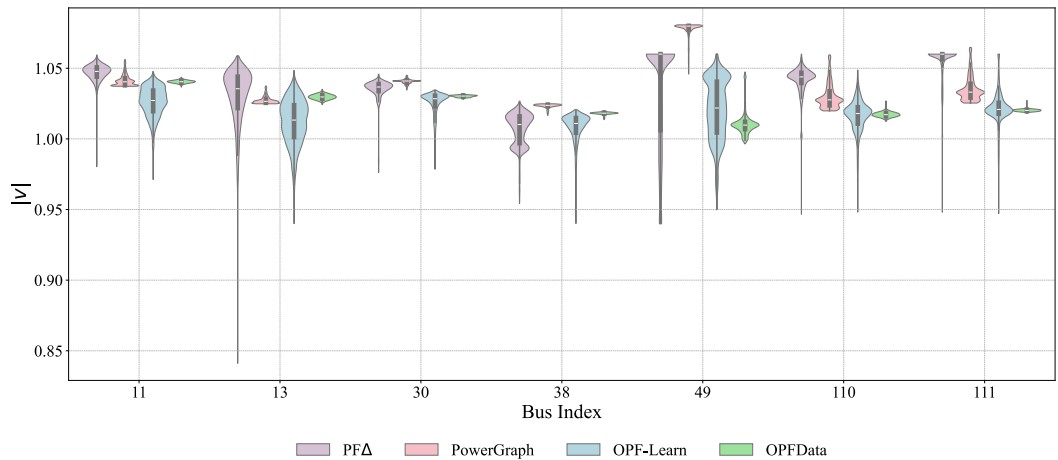

Figure A.7: Violin plots illustrating spread of $|v|$ values in 7 randomly selected buses. This graphic compares feature diversity of $|v|$ values sampled from $PF\Delta$ to other large-scale benchmark datasets.

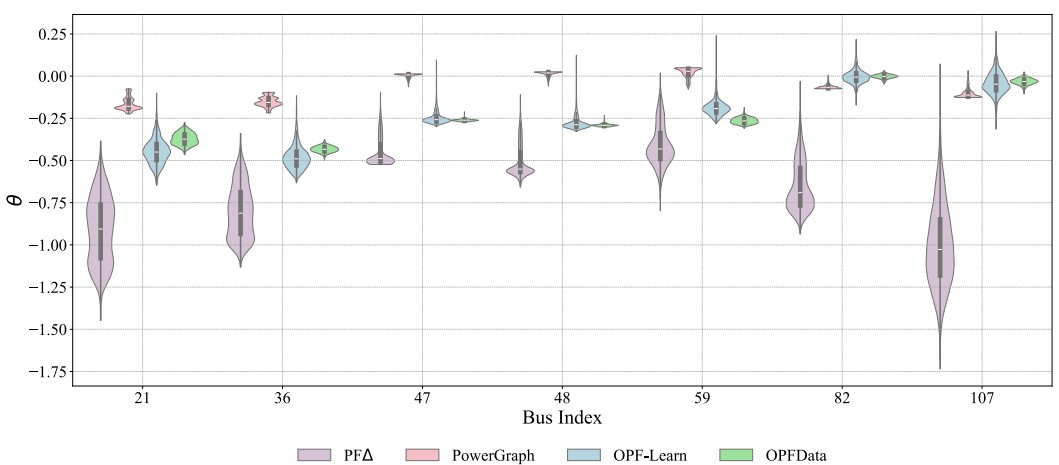

Figure A.8: Violin plots illustrating spread of $\theta$ values in 7 randomly selected buses. This graphic compares feature diversity of $\theta$ values sampled from $PF\Delta$ to other large-scale benchmark datasets.

## A.5 Summary of Dataset Splits

Tables A.2 and A.3 summarize the breakdown of data that **PF**$\Delta$ provides for each bus system size. Specifically, the tables illustrate how the data is distributed across each feasibility regime, topological perturbation type, and training and testing splits. Each bus system size contains a total of 132,000 data samples with the exception of the GOC-2000 system, which only contains 30,000 samples. We note that the data for GOC-2000 system can only be used for Task 1.3 (described in Table 2). We also note that we never test on the Approaching Infeasible regime, as this subset of data is only used for data augmentation in Task 4.2 to assess whether including this subset improves testing performance on Close-to-Infeasible samples.

| Bus System Size | Feasible | | | Approaching Infeasible | | | Close-to-Infeasible | | |
|---|---|---|---|---|---|---|---|---|---|
| | N | N-1 | N-2 | N | N-1 | N-2 | N | N-1 | N-2 |
| IEEE-14 | 54,000 | 27,000 | 18,000 | 14,400 | 14,400 | 14,400 | 3,600 | 3,600 | 3,600 |
| IEEE-30 | 54,000 | 27,000 | 18,000 | 14,400 | 14,400 | 14,400 | 3,600 | 3,600 | 3,600 |
| IEEE-57 | 54,000 | 27,000 | 18,000 | 14,400 | 14,400 | 14,400 | 3,600 | 3,600 | 3,600 |
| IEEE-118 | 54,000 | 27,000 | 18,000 | 14,400 | 14,400 | 14,400 | 3,600 | 3,600 | 3,600 |
| GOC-500 | 54,000 | 27,000 | 18,000 | 14,400 | 14,400 | 14,400 | 3,600 | 3,600 | 3,600 |
| GOC-2000 | 27,000 | 13,500 | 9,000 | 2,400 | 2,400 | 2,400 | 600 | 600 | 600 |

Table A.2: Number of available *training* datapoints provided by **PF**Δ across bus system sizes, feasibility regimes, and grids with varying topological perturbations.

| Bus System Size | Feasible | | | Approaching Infeasible | | | Close-to-Infeasible | | |
|---|---|---|---|---|---|---|---|---|---|
| | N | N-1 | N-2 | N | N-1 | N-2 | N | N-1 | N-2 |
| IEEE-14 | 2,000 | 2,000 | 2,000 | - | - | - | 200 | 200 | 200 |
| IEEE-30 | 2,000 | 2,000 | 2,000 | - | - | - | 200 | 200 | 200 |
| IEEE-57 | 2,000 | 2,000 | 2,000 | - | - | - | 200 | 200 | 200 |
| IEEE-118 | 2,000 | 2,000 | 2,000 | - | - | - | 200 | 200 | 200 |
| GOC-500 | 2,000 | 2,000 | 2,000 | - | - | - | 200 | 200 | 200 |
| GOC-2000 | 1,000 | 1,000 | 1,000 | - | - | - | 100 | 100 | 100 |

Table A.3: Number of available *testing* datapoints provided by **PF**Δ across bus system sizes, feasibility regimes, and grids with varying topological perturbations.

## A.6 PyTorch InMemoryDataset Format of PFΔ

We provide a PyTorch `InMemoryDataset` class called `PFDeltaDataset` to load the raw data described in Appendix A.5. This dataset class is designed to support loading data for a specified task for with a given bus system size. Each raw data instance that we load is stored as a JSON file representing the power flow solution for a specific grid configuration.

We offer two different `HeteroData` formulations of the dataset:

- **Component-Level Formulation.** This representation includes nodes corresponding to physical components such as buses, generators, and loads. It includes both nodal and edge attributes, all expressed in per-unit (p.u.) values. The structure of this formulation follows Figure A.9.

- **PV-PQ Formulation.** Optionally-, users can use this formulation, which identifies each bus as a PV, PQ, or slack bus. This formulation is tailored to models that use unique high-dimensional projections for PV and PQ buses. This formulation contains extra edge connections between bus nodes and their specific types (PV, PQ, or slack). The structure of this formulation follows Figure A.10. This formulation can be accessed by setting the `add_bus_type` flag to `True` in the `PFDeltaDataset` class.

Custom `PFDeltaDataset` classes can be implemented for each model by inheriting the base `PFDeltaDataset` class and overriding the `build_heterodata` method to perform model-specific data pre-processing of the `HeteroData` objects.

```
HeteroData(
  bus={
    x=[14, 2],
    y=[14, 2],
    bus_gen=[14, 2],
    bus_demand=[14, 2],
    bus_voltages=[14, 2],
    bus_type=[14],
    shunt=[14, 2],
    limits=[14, 2],
  },
  gen={
    limits=[5, 4],
    generation=[5, 2],
    slack_gen=[5],
  },
  load={ demand=[11, 2] },
  (bus, branch, bus)={
    edge_index=[2, 20],
    edge_attr=[20, 8],
    edge_label=[20, 4],
    edge_limits=[20, 1],
  },
  (gen, gen_link, bus)={ edge_index=[2, 5] },
  (bus, gen_link, gen)={ edge_index=[2, 5] },
  (load, load_link, bus)={ edge_index=[2, 11] },
  (bus, load_link, load)={ edge_index=[2, 11] }
)
```

Figure A.9: Component-level formulation of a `HeteroData` instance of the IEEE-14 bus system available on `PFDeltaDataset`.

Each `HeteroData` instance contains nodal and edge attributes that have been preprocessed from the dataset's raw power flow solutions. Edges represent electrical or logical connections between components and buses. Each such edge includes an `edge_index` tensor that defines graph connectivity using the COO (coordinate) format, with two tensors indicating the source and destination nodes of each edge type in the graph.

Nodes contain several types of attributes that can be utilized during the GNN message passing:

- **bus**
  - **x**: Input features for all buses. This is the active and reactive power demand $(p_d, q_d)$ for PQ buses, the net active power and voltage magnitude $(p_{net}, |v|)$ for PV buses, and the voltage angle and magnitude $(\theta, |v|)$ for the slack bus.
  - **y**: Output targets for all buses. Voltage angles and magnitudes $(\theta, |v|)$ for PQ buses, the net reactive power and voltage angles $(q_{net}, |v|)$ for the PV buses, and the net active and reactive powers $(p_{net}, q_{net})$ for the slack bus.
  - **bus_gen**: Active and reactive power generation at each bus $(p_g, q_g)$.
  - **bus_demand**: Active and reactive power demand at each bus $(p_d, q_d)$.
  - **bus_voltages**: Voltage angle and magnitude at each bus $(\theta, |v|)$.
  - **bus_type**: Integer type flag (1 = PQ, 2 = PV, 3 = slack)
  - **bus_shunt**: Shunt conductance and susceptance (the real and imaginary components of shunt admittances) $(g_s, b_s)$
  - **limits**: Voltage limits in the original pglib case. Note that only certain voltage limits are enforces, as specified by A.3.

```
HeteroData(
  bus={
    x=[14, 2],
    y=[14, 2],
    bus_gen=[14, 2],
    bus_demand=[14, 2],
    bus_voltages=[14, 2],
    bus_type=[14],
    shunt=[14, 2],
    limits=[14, 2],
  },
  gen={
    limits=[5, 4],
    generation=[5, 2],
    slack_gen=[5],
  },
  load={ demand=[11, 2] },
  PQ={
    x=[9, 2],
    y=[9, 2],
  },
  PV={
    x=[4, 2],
    y=[4, 2],
    generation=[4, 2],
    demand=[4, 2],
  },
  slack={
    x=[1, 2],
    y=[1, 2],
    generation=[1, 2],
    demand=[1, 2],
  },
  (bus, branch, bus)={
    edge_index=[2, 20],
    edge_attr=[20, 8],
    edge_label=[20, 4],
    edge_limits=[20, 1],
  },
  (gen, gen_link, bus)={ edge_index=[2, 5] },
  (bus, gen_link, gen)={ edge_index=[2, 5] },
  (load, load_link, bus)={ edge_index=[2, 11] },
  (bus, load_link, load)={ edge_index=[2, 11] },
  (PV, PV_link, bus)={ edge_index=[2, 4] },
  (bus, PV_link, PV)={ edge_index=[2, 4] },
  (PQ, PQ_link, bus)={ edge_index=[2, 9] },
  (bus, PQ_link, PQ)={ edge_index=[2, 9] },
  (slack, slack_link, bus)={ edge_index=[2, 1] },
  (bus, slack_link, slack)={ edge_index=[2, 1] }
)
```

Figure A.10: PV-PQ-level formulation of a `HeteroData` instance of the IEEE-14 bus system available on `PFDeltaDataset`. This formulation adds extra sub-node connections to the bus nodes to indicate whether they are PV, PQ, or slack nodes.

- **gen**
    - **limits**: Generator operating limits $(p_{\min}, p_{\max}, q_{\min}, q_{\max})$ as specified in the original pglib case. Note that only certain generation limits are enforced, as specified by A.3.
    - **generation**: Active and reactive power generation at generators $(p_g, q_g)$
    - **slack_gen**: Boolean indicator for whether the generator is located at a slack bus.
- **load**
    - **demand**: Active and reactive power demand at each load $(p_d, q_d)$.
- **PQ**
    - **x**: Active and reactive power demand at the bus $(p_d, q_d)$.
    - **y**: Voltage angles and magnitudes at the bus $(\theta, |v|)$.
- **PV**
    - **x**: Net active power and voltage magnitude at the bus $(p_{\text{net}}, |v|)$.
    - **y**: Net reactive power and voltage angle at the bus $(q_{\text{net}}, |v|)$
    - **generation**: Active and reactive power generation at the generators connected to the bus $(p_g, q_g)$.
    - **demand**: Active and reactive power demand at the load connected to the bus $(p_d, q_d)$.
- **slack**
    - **x**: Voltage angle and magnitude $(\theta, |v|)$.
    - **y**: Net active and reactive powers $(p_{\text{net}}, q_{\text{net}})$.
    - **generation**: Active and reactive power generation at the generators connected to the bus $(p_g, q_g)$.
    - **demand**: Active and reactive power demand at the load connected to the bus $(p_d, q_d)$.
- **(bus, branch, bus)**
    - **edge_attr**: Electrical branch characteristics: $(r, x, g_{\text{from}}, b_{\text{from}}, g_{\text{to}}, b_{\text{to}}, \tau, \theta_{\text{shift}})$
        * $r, x$: Series resistance and reactance
        * $g_{\text{from}}, b_{\text{from}}$: Shunt conductance and susceptance at the "from" bus
        * $g_{\text{to}}, b_{\text{to}}$: Shunt conductance and susceptance at the "to" bus
        * `tap` ($\tau$): Tap ratio (defaults to 1.0 if no transformer)
        * `shift` ($\theta_{\text{shift}}$): Phase shift angle (degrees)
    - **edge_label**: Power flow targets for each branch $(p_{\text{from}}, q_{\text{from}}, p_{\text{to}}, q_{\text{to}})$
        * $p_{\text{from}}, q_{\text{from}}$: active/reactive power at the source of the edge.
        * $p_{\text{to}}, q_{\text{to}}$: active/reactive power at the destination of the edge.
    - **edge_limits**: Branch flow limits as specified in the original pglib case. Note that these limits are not enforced in our data generation process, but are included in the dataset for analysis purposes.

### A.7 Interpretability Metrics

In addition to model performance metrics, we also include interpretability metrics to perform qualitative analysis on the solutions predicted by the ML-based power flow solvers. These metrics try to quantify the central tendencies and ranges of the predicted values of the model, such as voltage magnitudes, reactive powers, voltage phase differences, and power at the slack bus. These metrics are as follows:

**Voltage Magnitude at PQ Buses (Mean):** Calculate the mean voltage magnitude at PQ buses for a given sample.

$$\text{Mean}_{\text{sample}}(|v|) = \frac{1}{N} \sum_{i=1}^{N} |v|_i \tag{A.4}$$

where $N$ is the number of PQ buses in the sample, and $|v|_i$ is the voltage magnitude at the $i$-th bus. Aggregate for a dataset by calculating the mean and standard deviation across all samples.

**Voltage Magnitude at PQ Buses (Min):** Calculate the minimum voltage magnitude at PQ buses for a given sample.

$$\text{Min}_{\text{sample}}(|v|) = \min_{i=1}^{N} |v|_i \tag{A.5}$$

where $N$ is the number of PQ buses in the sample, and $|v|_i$ is the voltage magnitude at the $i$-th bus. Aggregate for a dataset by calculating the minimum across all samples.

**Voltage Magnitude at PQ Buses (Max):** Calculate the maximum voltage magnitude at PQ buses for a given sample.

$$\text{Max}_{\text{sample}}(|v|) = \max_{i=1}^{N} |v|_i \tag{A.6}$$

where $N$ is the number of PQ buses in the sample, and $|v|_i$ is the voltage magnitude at the $i$-th bus. Aggregate for a dataset by calculating the maximum across all samples.

**Reactive Power at PV Buses (Mean):** Calculate the mean reactive power at PV buses for a given sample.

$$\text{Mean}_{\text{sample}}(q_{\text{net}}) = \frac{1}{N} \sum_{i=1}^{N} q_{\text{net},i} \tag{A.7}$$

where $N$ is the number of PV buses in the sample, and $q_{\text{net},i}$ is the reactive power at the $i$-th bus. Aggregate for a dataset by calculating the mean and standard deviation across all samples.

**Reactive Power at PV Buses (Min):** Calculate the minimum reactive power at PV buses for a given sample.

$$\text{Min}_{\text{sample}}(q_{\text{net}}) = \min_{i=1}^{N} q_{\text{net},i} \tag{A.8}$$

where $N$ is the number of PV buses in the sample, and $q_{\text{net},i}$ is the reactive power at the $i$-th bus. Aggregate for a dataset by calculating the minimum across all samples.

**Reactive Power at PV Buses (Max):** Calculate the maximum reactive power at PV buses for a given sample.

$$\text{Max}_{\text{sample}}(q_{\text{net}}) = \max_{i=1}^{N} q_{\text{net},i} \tag{A.9}$$

where $N$ is the number of PV buses in the sample, and $q_{\text{net},i}$ is the reactive power at the $i$-th bus. Aggregate for a dataset by calculating the maximum across all samples.

**Voltage Phase Difference at All Branches (Mean):** Calculate the mean absolute voltage phase difference at all branches for a given sample.

$$\text{Mean}_{\text{sample}}(|\Delta\theta|) = \frac{1}{N} \sum_{i=1}^{N} |\Delta\theta| \tag{A.10}$$

where $N$ is the number of branches in the sample, and $|\Delta\theta|$ is the voltage phase difference at the $i$-th branch. Aggregate for a dataset by calculating the mean and standard deviation across all samples.

**Voltage Phase Difference at All Branches (Min):** Calculate the minimum absolute voltage phase difference at all branches for a given sample.

$$\text{Min}_{\text{sample}}(|\Delta\theta|) = \min_{i=1}^{N} |\Delta\theta| \tag{A.11}$$

where $N$ is the number of branches in the sample, and $|\Delta\theta|$ is the voltage phase difference at the $i$-th branch. Aggregate for a dataset by calculating the minimum across all samples.

**Voltage Phase Difference at All Branches (Max):** Calculate the maximum absolute voltage phase difference at all branches for a given sample.

$$\text{Max}_{\text{sample}}(|\Delta\theta|) = \max_{i=1}^{N} |\Delta\theta| \tag{A.12}$$

where $N$ is the number of branches in the sample, and $|\Delta\theta|$ is the voltage phase difference at the $i$-th branch. Aggregate for a dataset by calculating the maximum across all samples.

**Active Power at the Slack Bus (Mean):** Calculate for a dataset by computing the mean and standard deviation of active powers of slack buses across all samples.

**Active Power at the Slack Bus (Min):** Calculate for a dataset by computing the minimum active power of slack buses across all samples.

**Active Power at the Slack Bus (Max):** Calculate for a dataset by computing the maximum active power of slack buses across all samples.

**Reactive Power at the Slack Bus (Mean):** Calculate for a dataset by computing the mean and standard deviation of reactive powers of slack buses across all samples.

**Reactive Power at the Slack Bus (Min):** Calculate for a dataset by computing the minimum reactive power of slack buses across all samples.

**Reactive Power at the Slack Bus (Max):** Calculate for a dataset by computing the maximum reactive power of slack buses across all samples.

In addition of reporting these metrics for our dataset, we also report them for the predictions of PowerFlowNet, GNS-S, and CANOS-PF when trained on IEEE-118 (Task 1.3) considering topology $N$ in tables A.4, A.5, and A.6.

| Task 1.3 (topology $N$) | PV | | | | | |
| --- | --- | --- | --- | --- | --- | --- |
| | $Q_{\text{net}}$ | | | $\theta$ | | |
| | Min | Mean | Max | Min | Mean | Max |
| Custom AC-OPF | -7.66 | 0.496 | 10.1 | -1.66 | -0.688 | 0.283 |
| CANOS-PF | -7.36 | 0.531 | 10.4 | -1.43 | -0.675 | 0.165 |
| GNS-S | -5.29 | 0.123 | 10.9 | -1.70 | -1.457 | -1.021 |
| PFNet | -7.69 | 0.479 | 10.2 | -1.70 | -0.696 | 0.266 |

Table A.4: Interpretability metrics for PV buses in Task 1.3 considering topology $N$.

| Task 1.3 (topology $N$) | PQ | | | | | |
| --- | --- | --- | --- | --- | --- | --- |
| | $\theta$ | | | $|V|$ | | |
| | Min | Mean | Max | Min | Mean | Max |
| Custom AC-OPF | -1.59 | -0.709 | 0.194 | 0.693 | 1.02 | 1.08 |
| CANOS-PF | -1.35 | -0.698 | 0.080 | 0.783 | 1.02 | 1.08 |
| GNS-S | -1.68 | -1.487 | -1.187 | 0.804 | 1.03 | 1.11 |
| PFNet | -1.63 | -0.716 | 0.201 | 0.740 | 1.02 | 1.08 |

Table A.5: Interpretability metrics for PQ buses in Task 1.3 considering topology $N$.

| Task 1.3 (topology $N$) | Slack | | | | | |
| --- | --- | --- | --- | --- | --- | --- |
| | $P_{\text{net}}$ | | | $Q_{\text{net}}$ | | |
| | Min | Mean | Max | Min | Mean | Max |
| Custom AC-OPF | 11.33 | 22.5 | 29.3 | -4.78 | -1.23 | 1.71 |
| CANOS-PF | 11.58 | 22.5 | 29.4 | -4.83 | -1.20 | 1.70 |
| GNS-S | -7.54 | 8.0 | 14.4 | -2.42 | -1.37 | 1.07 |
| PFNet | 12.63 | 22.7 | 28.7 | -5.71 | -1.21 | 1.60 |

Table A.6: Interpretability metrics for slack buses in Task 1.3 considering topology $N$.

## A.8 Replicating CANOS

To correctly compare the performance of CANOS-PF against that of PowerFlowNet and GNS-PF, we first re-implement the original CANOS to verify that our re-implementation works as expected. We assess the fidelity of our re-implementation by comparing its performance on its original dataset to the errors reported in its original paper [12]. Due to limited compute resources, we were unable to train CANOS using the hyperparameters specified in the main body of the original paper. Instead, we train a significantly smaller model for a smaller number of training steps. The results we report in this appendix are on CANOS with 16 message passing steps and hidden size of 256 trained on 200k training steps. In this appendix, we refer to this version of CANOS as Small CANOS.

One of the featured models in [12] is Wide CANOS, with 36 message passing steps and a hidden size 384 trained for 600k training steps. Table A.7 compares the MSE of different variables as well as their sum when trained on `pglib_opf_case500_goc`. As the table reveals, Small CANOS performs comparatively to Wide CANOS. While Small CANOS has a higher Total MSE, we attribute this discrepancy to the fact that Small CANOS is significantly smaller, and that Wide CANOS was trained for a significantly larger number of training steps. We thus conclude that our implementation of CANOS is accurate.

| Variable | Wide CANOS - Original, TopDrop | Small CANOS - Re-implementation |
|---|---|---|
| Total MSE | 1.63e-02 | 2.70e-02 |
| Bus VA | 1.59e-04 | 5.15e-04 |
| Bus VM | 1.45e-06 | 2.94e-05 |
| Gen Pg | 6.65e-04 | 2.01e-03 |
| Gen Qg | 1.79e-04 | 1.89e-03 |
| Line Pf | 2.02e-03 | 5.46e-03 |
| Line Pt | 2.01e-03 | 1.56e-03 |
| Line Qf | 4.43e-03 | 5.46e-03 |
| Line Qt | 3.99e-03 | 1.54e-03 |
| Transf. Pf | 1.22e-03 | 2.47e-03 |
| Transf. Pt | 1.22e-03 | 1.83e-03 |
| Transf. Qf | 2.21e-04 | 2.46e-03 |
| Transf. Qt | 2.23e-04 | 1.80e-03 |

Table A.7: MSE comparison of Wide CANOS (36 message passing steps, 384 hidden size, 600k training steps) and Small CANOS (16 message passing steps, 256 hidden size, 200k training steps).

## A.9 Replicating PowerFlowNet Results

To correctly compare the performance of PowerFlowNet against that of CANOS-PF and GNS-PF, we adapted the implementation of PowerFlowNet [7] provided by the authors. To verify that the model works as expected within our code repository, we retrain it on its original dataset using its own pre-processing. Table 2 in [7] reports a Masked L2 loss of 0.018 when trained on pglib_opf_case118_ieee. When trained in our code repository, we attained a Masked L2 loss of 0.0155. Thus, we conclude the model was correctly adapted to our code repository.

## A.10 Modified GraphNeuralSolver Formulation

In this section, we present the modified components of the GraphNeuralSolver (GNS) model formulation used in our experiments. Specifically, we highlight the differences from the original architecture proposed in [4], adapting it to accommodate a single slack bus and fixed active power generation at PV buses, as in traditional power flow formulations. For consistency, we retain the same notation as in [4], and only the equations that deviate from the original formulation are shown below.

$$p_{\text{Joule}}^k = \sum_{\substack{i:\text{``from''}\\j:\text{``to''}}} \Bigg| -v_i^k v_j^k y_{ij} \frac{1}{\tau_{ij}} \left( \cos(\theta_i - \theta_j - \delta_{ij} - \theta_{\text{shift},ij}) + \cos(\theta_j - \theta_i - \delta_{ij} + \theta_{\text{shift},ij}) \right)$$

$$+ \left( \frac{v_i^k}{\tau_{ij}} \right)^2 y_{ij} \cos(\delta_{ij}) + (v_j^k)^2 y_{ij} \cos(\delta_{ij}) \Bigg| \tag{A.13a}$$

$$p_{\text{global}}^k = \sum_{i=1}^{N} p_{d,i} + g_{s,i}(v_i^k)^2 + p_{\text{Joule}}^k \tag{A.13b}$$

$$\lambda^k = \begin{cases} \frac{p_{\text{global}}^k - \sum_{i \in \text{non-slack gen}} \mathring{p}_{g,i} - \overline{p}_{g,\text{slack}}}{2(\mathring{p}_{g,\text{slack}} - \underline{p}_{g,\text{slack}})}, & \text{if } p_{\text{global}}^k < \sum_{i \in \text{all gens}} \mathring{p}_{g,i} \\ \frac{p_{\text{global}}^k - \sum_{i \in \text{non-slack gen}} \mathring{p}_{g,i} - 2\mathring{p}_{g,\text{slack}} - \overline{p}_{g,\text{slack}}}{2(\overline{p}_{g,\text{slack}} - \mathring{p}_{g,\text{slack}})} & \text{otherwise} \end{cases} \tag{A.13c}$$

$$p_{g,i}^k = \begin{cases} p_{g,i}^k(\lambda^k), & \text{if } i \text{ is slack} \\ \mathring{p}_{g,i}, & \text{otherwise (keep original setpoint)} \end{cases} \tag{A.13d}$$

$$q_{g,i}^k = \left( q_{d,i} - b_{s,i}(v_i^k)^2 \right)$$

$$- \sum_{\substack{j \in \mathcal{N}(i)\\i:\text{``from''}}} \left[ +v_i^k v_j^k y_{ij} \frac{1}{\tau_{ij}} \cos(\theta_i - \theta_j - \delta_{ij} - \theta_{\text{shift},ij}) + \left( \frac{v_i^k}{\tau_{ij}} \right)^2 \left( y_{ij} \sin(\delta_{ij}) + \frac{b_{ij}}{2} \right) \right]$$

$$- \sum_{\substack{j \in \mathcal{N}(i)\\i:\text{``to''}}} \left[ +v_i^k v_j^k y_{ij} \frac{1}{\tau_{ij}} \sin(\theta_i - \theta_j - \delta_{ij} + \theta_{\text{shift},ij}) + (v_i^k)^2 \left( y_{ij} \sin(\delta_{ij}) + \frac{b_{ij}}{2} \right) \right]$$

$$\tag{A.13e}$$

$$\Delta p_i^k = \left( p_{g,i}^k - p_{d,i} - g_{s,i}(v_i^k)^2 \right)$$

$$+ \sum_{\substack{j \in \mathcal{N}(i)\\i:\text{``from''}}} \left[ -v_i^k v_j^k y_{ij} \frac{1}{\tau_{ij}} \cos(\theta_i - \theta_j - \delta_{ij} - \theta_{\text{shift},ij}) + \left( \frac{v_i^k}{\tau_{ij}} \right)^2 y_{ij} \cos(\delta_{ij}) \right]$$

$$+ \sum_{\substack{j \in \mathcal{N}(i)\\i:\text{``to''}}} \left[ -v_i^k v_j^k y_{ij} \frac{1}{\tau_{ij}} \cos(\theta_i - \theta_j - \delta_{ij} + \theta_{\text{shift},ij}) + (v_i^k)^2 y_{ij} \cos(\delta_{ij}) \right] \tag{A.13f}$$

$$\Delta q_i^k = \left( q_{g,i}^k - q_{d,i} + b_{s,i}(v_i^k)^2 \right)$$

$$+ \sum_{\substack{j \in \mathcal{N}(i)\\i:\text{``from''}}} \left[ -v_i^k v_j^k y_{ij} \frac{1}{\tau_{ij}} \sin(\theta_i - \theta_j - \delta_{ij} - \theta_{\text{shift},ij}) - \left( \frac{v_i^k}{\tau_{ij}} \right)^2 \left( y_{ij} \sin(\delta_{ij}) + \frac{b_{ij}}{2} \right) \right]$$

$$+ \sum_{\substack{j \in \mathcal{N}(i)\\i:\text{``to''}}} \left[ -v_i^k v_j^k y_{ij} \frac{1}{\tau_{ij}} \sin(\theta_i - \theta_j - \delta_{ij} + \theta_{\text{shift},ij}) - (v_i^k)^2 \left( y_{ij} \sin(\delta_{ij}) + \frac{b_{ij}}{2} \right) \right]$$

$$\tag{A.13g}$$

## A.11 Hyperparameter Tuning

We hyperparameter tuned each model on Task 1.3 to maximize generalizability. We employed a grid search strategy, sweeping multiple hyperparameters for each of the three models until convergence was achieved. The best hyperparameters were identified as the ones that performed the best on a validation set (this was designated as 10% of the training dataset) based on each model's native training loss. Each model was set to have a batch size of 64. Once an initial set of optimal hyperparameters were found, the learning rate of each model was fine-tuned based on the specific tasks we performed evaluation for (1.1, 1.2, 1.3, 2.3, 4.1, 4.2, and 4.3). This involved conducting a small sweep of learning rates on these specific tasks. Model-specific details on the hyperparameters tuned, the number of epochs trained for each model, and the final parameters are provided here:

- **PFNet**: We conducted a grid search over key hyperparameters, including `hidden_dim`, `n_gnn_layers`, `K` (the receptive field size in the TAGConv layers), and the learning rate. Each model was trained for approximately 100 epochs during tuning, consistent with the configuration in the original paper [7]. The final configuration used `hidden_dim` = 256, `n_gnn_layers` = 5, `K` = 4, and a dropout rate of 0.2. The learning rate was 0.0001 for Tasks 1.1, 1.2 and 4.3, 0.00009 for Task 1.3, and 0.0003 for Tasks 2.3, 4.1, and 4.2.

- **GNS-S**: We performed a grid search over the following hyperparameters: `K` (the depth of the network), `hidden_dim`, `gamma` (which weights the contribution of each iterative layer to the loss function), and the learning rate. The finalized values were: `K` = 10, `hidden_dim` = 20, and `gamma` = 0.01. The learning rate was 0.0003 for Tasks 1.1, 4.1, and 4.3, 0.0005 for Tasks 1.3 and 4.2, and 0.0007 for Tasks 1.2 and 2.3. The model was trained for 25 epochs, approximately consistent with the training setup in the original paper [4].

- **CANOS-PF**: We performed a grid search over the following hyperparameters: `hidden_dim`, `k_steps` (the depth of the message-passing network), and the learning rate. The learning rate scheduler was fixed to the same setup as the one defined in the original implementation and the model was trained for 50 epochs [12]. The finalized parameters were: `hidden_dim` = 128 and `k_steps` = 15. The learning rate was 0.0003 for Tasks 1.3, 4.1, and 4.2, 0.0005 for Tasks 1.1, 1.2, 4.3, and 0.0007 for Task 2.3.

## A.12 Extended Experimental Results

Tables A.8, A.9, A.10, A.11, and A.12 present extended experimental results on **PF$\Delta$**. Model performance is evaluated on Tasks 1.1, 1.2, 1.3 (and 2.1), 3.1, 4.1, 4.2, and 4.3, with analysis primarily focused on the IEEE 118-bus system. Power Balance Loss (PBL) mean and maximum values for the IEEE 118-bus system under Tasks 1.1, 1.3, 1.3/3.1, 4.1, 4.2, and 4.3 are reported in Tables A.8 and A.9. Additionally, PBL mean and maximum results across systems of varying size (IEEE-57, IEEE-118, and IEEE-500) for Task 3.1 are shown in Table A.10.

In addition to the three GNN-based models, we report results from the PowerModels.jl Newton–Raphson (NR) solver as a point of comparison. Runtimes for all four models across the three bus systems are provided in Table A.12. NR calculations were performed from a flat start, and results were averaged over the percentage of samples that converged. The convergence rates for the IEEE-57, IEEE-118, and IEEE-500 systems were 95.2%, 65.7%, and 43.4%, respectively. All reported results are obtained by training each model three times per experiment with randomly initialized weights, then computing the mean and standard deviation across these runs. Runtimes for NR were calculated on an Intel Xeon Gold 6140 CPU whereas runtimes for the GNN-based models were calculated on an NVIDIA RTX 8000 GPU.

While training, the performance of the model is calculated periodically in the validation set, which is set to be a fixed random subsample of 10% of the task's corresponding training set. Another key aspect of our training is the early stopping strategy we apply for each of the models. As described in Appendix A.11, each model requires a different number of epochs to reach convergence; to ensure that we are not overfitting, we employ an early stopping strategy that halts training if the epoch with the best validation PBL (Mean) happened over 15 epochs ago or if this same error has not changed by more than 1% for 10 epochs consecutively.

| Experiment | | Power Balance Loss (Mean) | | | |
|---|---|---|---|---|---|
| Task | Model | N | N-1 | N-2 | Close-to- infeasible |
| | PFNet | 3.2±0.2e-1 | 5.2±0.2e-1 | 5.3±0.3e-1 | 1.4±0.1e0 |
| 1.1 | CANOS-PF | 1.9±0.3e-1 | 7.6±0.8e-1 | 6.6±0.5e-1 | 1.2±0.1e0 |
| | GNS | 5.3±2.7e-1 | 6.8±2.6e-1 | 7.1±2.5e-1 | 1.0±0.2e0 |
| | PFNet | 3.2±0.2e-1 | 4.0±0.2e-1 | 4.4±0.2e-1 | 1.2±0.05e0 |
| 1.2 | CANOS-PF | 1.8±0.06e-1 | 2.2±0.08e-1 | 2.4±0.09e-1 | 8.7±0.7e-1 |
| | GNS | 3.6±0.3e-1 | 4.0±0.3e-1 | 4.1±0.3e-1 | 8.2±1.1e-1 |
| | PFNet | 3.4±0.08e-1 | 4.0±0.05e-1 | 4.2±0.09e-1 | 1.1±0.03e0 |
| 1.3, 2.1 | CANOS-PF | 1.9±0.2e-1 | 2.1±0.1e-1 | 2.2±0.1e-1 | 9.7±0.7e-1 |
| | GNS | 3.5±0.4e-1 | 3.8±0.4e-1 | 3.9±0.4e-1 | 7.7±0.5e-1 |
| | PFNet | 4.1±0.6e-1 | 4.9±0.7e-1 | 5.1±0.7e-1 | 1.1±0.08e0 |
| 2.3 | CANOS-PF | 4.1±0.8e-1 | 4.6±0.9e-1 | 4.7±0.8e-1 | 1.0±0.1e0 |
| | GNS | 8.3±2.6e-1 | 8.8±2.7e-1 | 8.9±2.7e-1 | 1.3±0.3e0 |
| | PFNet | 4.5±0.4e-1 | 5.2±0.09e-1 | 5.3±0.07e-1 | 1.3±0.04e0 |
| 4.1 | CANOS-PF | 4.7±0.1e-1 | 5.3±0.05e-1 | 5.4±0.06e-1 | 1.7±0.01e0 |
| | GNS | 3.5±0.1e-1 | 3.7±0.2e-1 | 3.8±0.2e-1 | 6.6±0.3e-1 |
| | PFNet | 5.6±0.7e-1 | 6.6±0.7e-1 | 6.7±0.7e-1 | 1.2±0.1e0 |
| 4.2 | CANOS-PF | 3.9±0.8e-1 | 4.3±0.8e-1 | 4.4±0.8e-1 | 8.8±0.9e-1 |
| | GNS | 4.0±0.9e-1 | 4.6±1.0e-1 | 4.6±1.0e-1 | 6.4±1.4e-1 |
| | PFNet | 1.2±0.04e0 | 1.4±0.04e0 | 1.5±0.02e0 | 1.1±0.07e0 |
| 4.3 | CANOS-PF | 1.1±0.04e0 | 1.2±0.03e0 | 1.2±0.02e0 | 0.8±0.05e0 |
| | GNS | 6.3±2.2e-1 | 7.3±2.4e-1 | 7.3±2.4e-1 | 7.0±2.8e-1 |

Table A.8: Power Balance Loss (Mean) across different grid conditions.

| Experiment | | Power Balance Loss (Max) | | | |
|---|---|---|---|---|---|
| Task | Model | N | N-1 | N-2 | Close-to- infeasible |
| | PFNet | 1.5±0.2e1 | 5.7±0.3e1 | 5.1±0.8e1 | 7.0±2.1e1 |
| 1.1 | CANOS-PF | 9.4±2.6e0 | 1.7±0.2e2 | 1.6±0.3e2 | 1.9±0.4e2 |
| | GNS | 2.3±1.0e1 | 8.9±4.2e1 | 1.4±0.7e2 | 8.4±5.6e1 |
| | PFNet | 1.1±0.1e1 | 4.2±0.7e1 | 5.0±1.3e1 | 3.7±0.6e1 |
| 1.2 | CANOS-PF | 3.9±1.1e0 | 1.2±0.06e1 | 1.5±0.1e1 | 2.7±0.4e1 |
| | GNS | 1.6±0.2e1 | 2.1±0.08e1 | 1.7±0.2e1 | 2.4±0.8e1 |
| | PFNet | 1.0±0.2e1 | 3.5±0.3e1 | 3.5±0.4e1 | 3.5±0.6e1 |
| 1.3, 2.1 | CANOS-PF | 3.7±0.1e0 | 1.1±0.1e1 | 8.9±2.1e0 | 3.4±0.4e1 |
| | GNS | 1.1±0.3e1 | 2.1±0.3e1 | 1.8±0.2e1 | 1.9±0.5e1 |
| | PFNet | 1.1±0.2e1 | 4.8±1.8e1 | 3.9±0.2e1 | 4.3±0.7e1 |
| 2.3 | CANOS-PF | 0.8±0.4e1 | 3.4±0.8e1 | 2.9±0.2e1 | 4.3±0.8e1 |
| | GNS | 3.7±3.5e1 | 5.4±5.1e1 | 5.3±3.8e1 | 4.0±2.3e1 |
| | PFNet | 1.5±0.2e1 | 4.5±0.2e1 | 4.3±0.3e1 | 8.5±0.8e1 |
| 4.1 | CANOS-PF | 1.1±0.08e1 | 3.2±0.2e1 | 3.1±0.2e1 | 9.1±0.8e1 |
| | GNS | 1.2±0.2e1 | 2.1±0.3e1 | 1.5±0.07e1 | 2.0±0.3e1 |
| | PFNet | 1.6±0.3e1 | 4.9±0.8e1 | 4.1±1.2e1 | 8.5±0.4e1 |
| 4.2 | CANOS-PF | 8.5±1.5e0 | 2.8±0.06e1 | 2.7±0.08e1 | 4.6±0.6e1 |
| | GNS | 1.8±0.6e1 | 2.4±0.3e1 | 2.5±1.4e1 | 2.2±0.2e1 |
| | PFNet | 3.3±0.8e1 | 6.0±2.2e1 | 5.0±0.9e1 | 8.1±0.3e1 |
| 4.3 | CANOS-PF | 2.0±0.5e1 | 4.0±0.3e1 | 5.2±0.3e1 | 2.4±0.3e1 |
| | GNS | 2.4±1.9e1 | 4.5±3.6e1 | 4.1±2.3e1 | 2.5±0.5e1 |

Table A.9: Power Balance Loss (Max) across different grid conditions.

| Experiment | | Power Balance Loss (Mean) | | | |
|---|---|---|---|---|---|
| Case | Model | N | N-1 | N-2 | Close-to-infeasible |
| 57 | PFNet | 2.3±0.4e0 | 2.3±0.4e0 | 2.3±0.4e0 | 2.4±0.4e0 |
| | CANOS-PF | 1.6±0.1e0 | 1.6±0.1e0 | 1.6±0.1e0 | 1.7±0.1e0 |
| | GNS | 3.3±1.4e1 | 8.8±3.9e1 | 1.5±0.7e2 | 0.8±0.2e0 |
| | NR | 1.1±0.0e-6 | 1.2±0.0e-6 | 1.1±0.0e-6 | 1.3±0.0e-6 |
| 118 | PFNet | 3.4±0.08e-1 | 4.0±0.05e-1 | 4.2±0.09e-1 | 1.1±0.03e0 |
| | CANOS-PF | 1.9±0.2e-1 | 2.1±0.1e-1 | 2.2±0.1e-1 | 9.7±0.7e-1 |
| | GNS | 3.5±0.4e-1 | 3.8±0.4e-1 | 3.9±0.4e-1 | 7.7±0.5e-1 |
| | NR | 3.7±0.0e-6 | 3.2±0.0e-6 | 3.3±0.0e-6 | 4.6±0.0e-6 |
| 500 | PFNet | 8.3±3.9e1 | 8.1±3.8e1 | 8.4±4.0e1 | 8.9±3.8e1 |
| | CANOS-PF | 1.8±0.1e1 | 1.8±1.00e1 | 1.8±0.1e1 | 1.9±0.1e1 |
| | GNS | 2.4±0.7e1 | 2.4±0.7e1 | 2.4±0.7e1 | 2.4±0.7e1 |
| | NR | 1.4±0.0e-5 | 1.4±0.0e-5 | 1.3±0.0e-5 | 1.6±0.0e-5 |

Table A.10: Power Balance Loss (Mean) across different bus sizes and grid conditions.

| Experiment | | Power Balance Loss (Max) | | | |
|---|---|---|---|---|---|
| Case | Model | N | N-1 | N-2 | Close-to-infeasible |
| 57 | PFNet | 1.2±0.2e1 | 1.6±0.2e1 | 1.5±0.2e1 | 2.1±0.2e1 |
| | CANOS-PF | 3.7±0.4e1 | 3.9±0.2e1 | 3.8±0.3e1 | 3.9±0.2e1 |
| | GNS | 3.0±1.3e5 | 4.4±2.2e5 | 5.2±2.4e5 | 4.3±1.8e1 |
| 118 | PFNet | 1.0±0.2e1 | 3.5±0.3e1 | 3.5±0.4e1 | 3.5±0.6e1 |
| | CANOS-PF | 3.7±0.1e0 | 1.1±0.1e1 | 8.9±2.1e0 | 3.4±0.4e1 |
| | GNS | 1.1±0.3e1 | 2.1±0.3e1 | 1.8±0.2e1 | 1.9±0.5e1 |
| 500 | PFNet | 1.9±0.6e3 | 1.9±0.6e3 | 1.9±0.6e3 | 1.9±0.6e3 |
| | CANOS-PF | 4.7±0.5e2 | 5.1±0.4e2 | 6.1±0.2e2 | 4.9±0.7e2 |
| | GNS | 5.9±2.3e2 | 5.9±2.3e2 | 6.1±2.2e2 | 5.8±2.1e2 |

Table A.11: Power Balance Loss (Max) across different bus sizes and grid conditions.

| Experiment | | Runtime (seconds) | | | |
|---|---|---|---|---|---|
| Case | Model | N | N-1 | N-2 | Close-to-infeasible |
| 57 | PFNet | 6.6e-3±2.8e-5 | 6.6e-3±1.6e-5 | 6.6e-3±2.8e-5 | 6.7e-3±8.5e-5 |
| | CANOS-PF | 3.1e-2±8.1e-5 | 3.1e-2±2.1e-5 | 3.1e-2±1.1e-4 | 3.2e-2±9.8e-5 |
| | GNS | 5.1e-2±3.1e-4 | 5.1e-2±1.5e-4 | 5.1e-2±9.6e-5 | 5.3e-2±1.0e-4 |
| | NR | 9.5e-3±2.0e-4 | 3.9e-3±8.4e-5 | 3.7e-3±4.1e-5 | 3.9e-3±1.4e-4 |
| 118 | PFNet | 6.4e-3±7.7e-5 | 6.4e-3±6.3e-5 | 6.4e-3±2.8e-5 | 6.4e-3±1.3e-5 |
| | CANOS-PF | 2.7e-2±2.1e-4 | 2.7e-2±2.1e-5 | 2.7e-2±8.8e-5 | 2.8e-2±1.8e-4 |
| | GNS | 4.8e-2±1.9e-4 | 4.8e-2±2.1e-4 | 4.8e-2±1.8e-4 | 4.9e-2±6.8e-5 |
| | NR | 4.2e-2±5.5e-4 | 1.6e-2±8.5e-4 | 1.2e-2±1.8e-4 | 1.2e-2±3.1e-4 |
| 500 | PFNet | 7.4e-3±1.5e-4 | 7.3e-3±4.0e-6 | 7.3e-3±2.3e-5 | 7.5e-3±5.0e-5 |
| | CANOS-PF | 3.1e-2±1.2e-4 | 3.1e-2±1.4e-4 | 3.1e-2±5.6e-4 | 3.2e-2±5.3e-4 |
| | GNS | 6.5e-2±1.6e-4 | 6.5e-2±2.3e-4 | 6.5e-2±3.0e-4 | 6.7e-2±5.0e-4 |
| | NR | 1.2e-1±3.1e-3 | 5.2e-2±2.4e-3 | 2.1e-2±1.1e-3 | 2.1e-2±1.0e-4 |

Table A.12: Runtimes across different bus sizes and grid conditions.

## A.13  Failure Modes of Models

Figure A.11 shows histograms of the maximum power balance loss (PBL Max) across all test samples, separated by model (*CANOS*, *PFNet*, *GNS-S*). Each model exhibits a left-skewed distribution with a long tail, indicating that while most samples have relatively low PBL Max values, a small subset experiences substantially larger violations. The spread and shape of these distributions differ across models: *CANOS* displays the tightest distribution with the lowest mean and standard deviation,

suggesting greater consistency and overall accuracy. *PFNet* exhibits a broader spread and a slightly higher mean, whereas *GNS-S* shows the widest spread.

Figures A.12-A.14 show histograms of power loss (PBL) per node, categorized by bus type (PQ, PV, Slack) for each model. Across all models, PQ buses consistently have lower PBL than PV buses. Some possible interpretations of this behavior include: (1) reactive power (predicted at PV buses) is harder to predict correctly than voltage magnitude (predicted at PQ buses), or (2) errors in reactive power contribute more heavily to overall PBL than errors in voltage magnitude. Performance on the slack bus varies distinctly across models. *CANOS-PF* achieves exactly zero PBL on the slack bus by design, due to its analytical treatment of that node. It also has the lowest PV and PQ PBL values in comparison to the other models. In contrast, *PFNet* and *GNS-S* display broad distributions of slack bus PBL, indicating greater variability and potential model instability at the slack node.

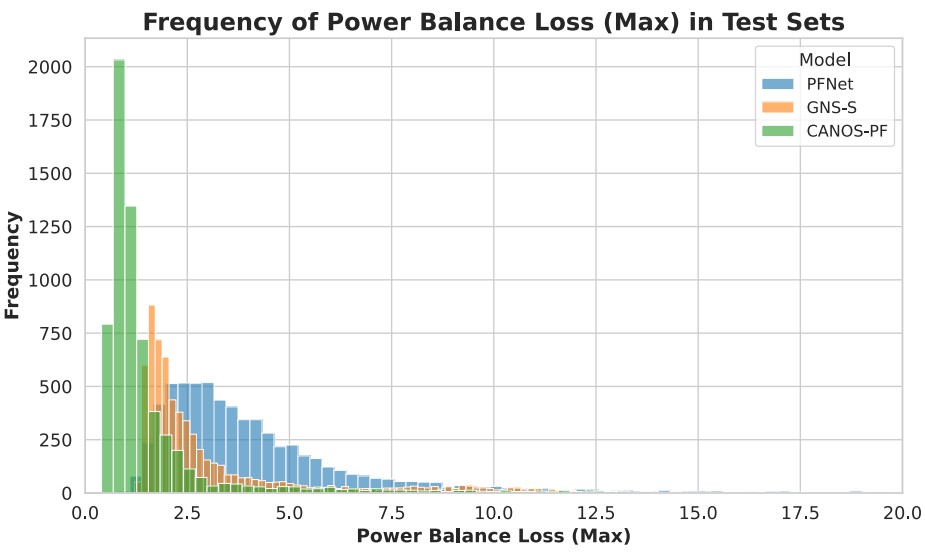

Figure A.11: Distribution of maximum power balance loss (PBL Max) across test samples for CANOS, PFNet, and GNS-S.

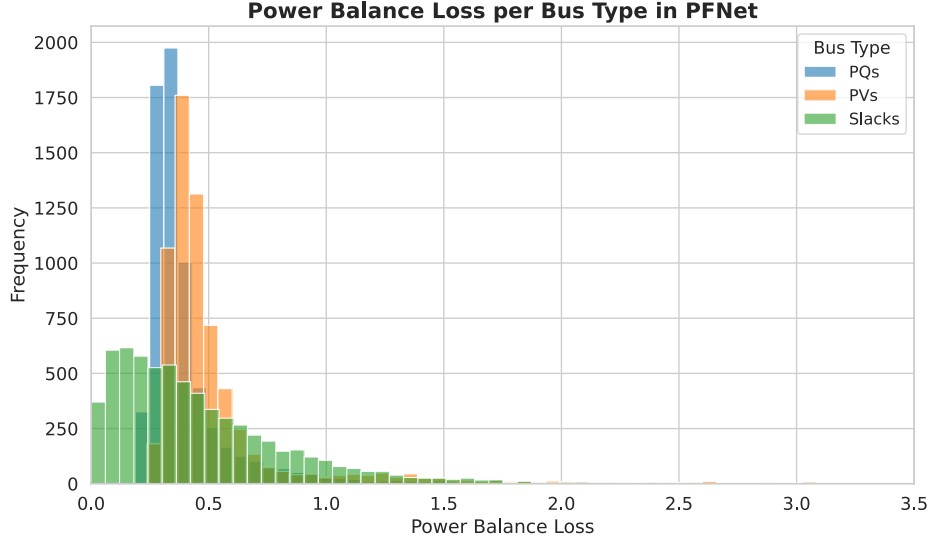

Figure A.12: Distribution of power balance loss (PBL Mean) across bus types within the test samples for PFNet.

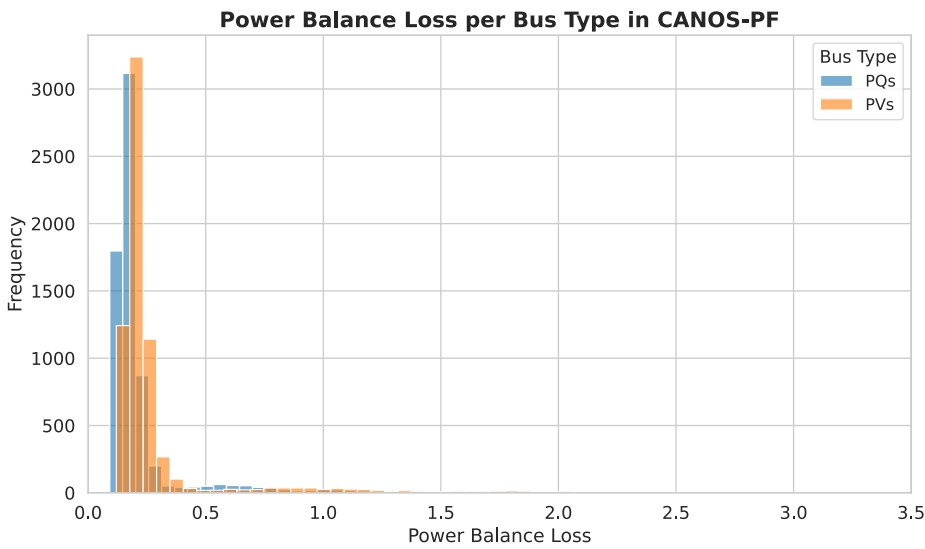

Figure A.13: Distribution of power balance loss (PBL Mean) across bus types within the test samples for CANOS-PF. All slack buses achieve a PBL of zero due to analytical enforcement of this property within the model. As a result, we do not present these results in this plot.

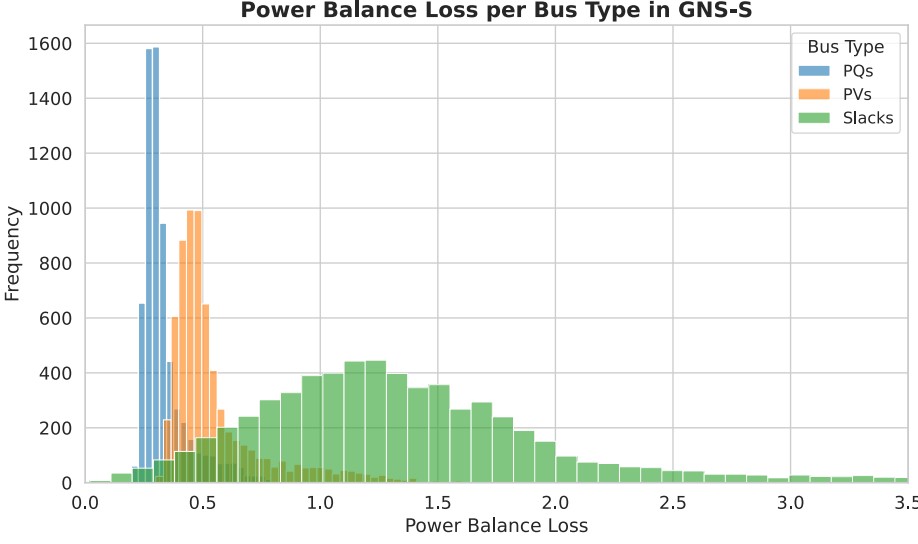

Figure A.14: Distribution of power balance loss (PBL Mean) across bus types within the test samples for GNS-S.

