# OpenReview forum: "PF∆: A Benchmark Dataset for Power Flow under Load, Generation, and Topology Variations"
_NeurIPS.cc/2025/Datasets_and_Benchmarks_Track — NeurIPS 2025 Datasets and Benchmarks Track poster_

### Official Review · Reviewer_AtCg · 2025-06-18

**Rating:** 4
**Confidence:** 4

**Summary:**

The paper introduces a benchmark data set for power flow analysis. Notably, it contains various variations, such as load and topology variations.

**Dataset Code Accessibility:**

Yes

**Ethical Considerations:**

No, there are no or only very minor ethics concerns

**Final Justification:**

The author's rebuttal addressed my major concerns, and I therefore already increased my score. Another reviewer criticised the lack of realism. While their criticism sounds reasonable to me, I am unable to assess this aspect due to limited domain knowledge on my side. I am therefore unable to champion this paper.

**Limitations Weaknesses:**

Some technical jargon must be better explained, and the provided Python code and GitHub repository require a revision as well.

For instance, please explain the meaning of "N-1/N-2 topological perturbations" when first introduced in the paper. It becomes somewhat clear in lines 134, but the term was used a couple of times before that.

Some comments on the linked GitHub repository:
- Please specify a license (maybe the same as used on the HuggingFace record)
- Please provide some instructions on how to run the code (i.e., what Python version is required, etc.)
- Please describe the usage (incl. documentation) of the PyTorch InMemoryDataset class in the README or link to some documentation -- similar to what is described in A.6
- Consider using type hints
- I do not understand the directory structure described in the README. Where can I find the .tar.gz archives? Do I have to generate them first, or can I download them from Hugging Face? Please make this clearer in the README.
- Why is the FashionMNIST data set included? (see core/datasets/fashionmnist.py)
- Would be nice to provide links/references in the README for the models (e.g., CANOS, GNS, PFNet) and other software packages such as PowerModels.jl

Minor:
- Turn Figure 1-3 into vector graphics (also holds for the Figures in the appendix)
- Figure A.2: Caption is cut off

**Strengths Contributions:**

Overall, the paper is well written and easy to follow. Furthermore, it provides a comprehensive evaluation of state-of-the-art models on the proposed benchmark. The benchmark dataset itself is novel. In particular, it contains various variations, such as load and topology variations, making it interesting for researchers applying AI/ML to power flow prediction.

---

> ### Author Rebuttal · Authors · 2025-07-30
>
> Thank you for your careful and thoughtful review. We would like to address the limitations raised by the reviewer:
>
> > Some technical jargon must be better explained… For instance, please explain the meaning of "N-1/N-2 topological perturbations" when first introduced in the paper. It becomes somewhat clear in lines 134, but the term was used a couple of times before that.
>
> Thank you for bringing this oversight to our attention. An N-k topological perturbation refers to the simultaneous outage or loss of k system components, typically generators and/or transmission lines. We will revise the manuscript to ensure this definition is provided clearly at the first mention of the term, and more generally to reduce and define any jargon.
>
> > Please specify a license (maybe the same as used on the HuggingFace record)
>
> Thank you for bringing this oversight to our attention. We will include an MIT License once we’ve made the repository public on Github.
>
> > Please provide some instructions on how to run the code (i.e., what Python version is required, etc.)
>
> > Please describe the usage (incl. documentation) of the PyTorch InMemoryDataset class in the README or link to some documentation -- similar to what is described in A.6
>
> Thank you for this comment. We plan to significantly expand the instructions on the README and documentation of the code repository. This includes the following (1) We will transform the file requirements.txt to a pipwheel to facilitate the installation of the required packages, (2) We will add a notebook that gives examples of how to access and use the PFDelta data, (3) We will include thorough docstrings for every major class.
>
> To illustrate some of the changes we plan to implement, we include: (1) a snippet from the updated README that provides an overview of the PFDeltaDataset class and points to more detailed usage instructions (available on a Jupyter notebook that we will include in the updated repository), as well as how to replicate results from the paper using the codebase and (2) an example docstring to facilitate easier understanding of how each function in the codebase works.
>
> README SNIPPET:
>
> How to use PFDelta
> --------------------------------
>
> ### Code requirements
>
>  The Python packages needed to use PFDelta and to replicate the results of the paper are listed in the file `requirements.txt`. They can be installed using the command:
>
>  ```bash
>  pip install -r requirements.txt
>  ```
>  The Julia packages needed to use the proposed data generation framework are listed in `data_generation/Project.toml`. They can be installed using the command:
>  ```bash
>  julia --project=data_generation -e 'using Pkg; Pkg.instantiate()'
>  ```
>
> ### Using the data
>
> We provide a PyTorch dataset class to download and preprocess the raw data stored in HuggingFace. This dataset class can be prompted to load the train/val/test set for a given task in the benchmark and it preprocesses the data to enable both supervised and unsupervised methods. Each dataset item is preprocessed as a graph that contains the sufficient network data to run a standard Newton-Raphson algorithm, and to calculate Power Balance Loss, as well as a single solution that can be used as ground truth by supervised losses. An in-depth description of the data structure is located in Appendix A.6 of the PFDelta paper, and is replicated below:
>
> (...)
>
> The dataset class is located in `core/datasets/pfdelta_dataset.py`, saved under the name PFDeltaDataset. A thorough description of how to use the dataset class is included in the docstrings of the class. In addition to this, we have included a notebook with examples of how to use the dataset class. This includes an example for how to modify the class to adapt the preprocessing to different models’ needs, such as homogeneous GNNs and standard feedforward networks.
>
> ### Replicating results
>
> To replicate the results in the paper, use the config files located in `core/configs`. In particular, this folder contains one YAML file for each model and task task, named as `core/configs/{modelname}_task_{tasknumber}.yaml,` where {modelname} can be [list] and {tasknumber} can be [list]. For example, running the following command will retrain PowerFlowNet on Task 1.3:
>
> ```bash
> python main.py --config pfnet_task_1_3
> ```
>
> Additionally, the config file that verifyies the correct re-implementation of CANOS is also included. Running the command will create the folder `runs` that will contain the results -- e.g. `runs/canos_pf_task_1_1_<YYMMDD_HHMMSS>`. This folder has the following structure.
> ```
> runs/
> └── modelname_task_#_#_<YYMMDD_HHMMSS>
>     ├── config.yaml         # Model and training hyperparameters
>     ├── model.pt            # Trainable weights of the model
>     ├── out.txt             # Console output
>     ├── summary.json        # Training and validation losses summary
>     ├── train.json          # Training loss esper epoch
>     ├── val.json            # Validation loss every 2 epochs
> ```
> The `summary.json` file can be quickly used to compare the model's performance against the values stated in the paper. Note that our results are the test set errors averaged over 3 different RNG seeds, while the losses reported in `summary.json` will be on the validation set over a single RNG seed. This can create some discrepancies when comparing to the paper's values.
>
> EXAMPLE DOCSTRING:
>
> ```python
> class PowerBalanceLoss:
>    """
>    Computes the power balance loss.
>
>    Args:
>        model (str): String identifier for the model type (e.g., "CANOS", "PFNet", or "GNS"). This is used internally to correctly parse and interpret specific model outputs.
>
>    Call Args:
>        outputs: Model outputs containing bus predictions (voltages, angles, power injections, etc.).
>        data (HeteroData): Graph data object containing bus, branch, and generator information, along with connectivity and attributes.
>
>    Returns:
>        torch.Tensor: The mean power balance loss (PBL Mean), computed as the average magnitude of squared active and reactive power mismatches across all buses.
>    """
>
>    def __init__(self, model):
>        self.power_balance_mean = None
>        self.power_balance_max = None
>        self.power_balance_l2 = None
>        self.model = model
>        self.loss_name = "PBL Mean"
>
>    def __call__(self, outputs, data: HeteroData):
> ```
> > Consider using type hints
>
> Thank you for bringing this oversight to our attention. We will make use of a linter to better format the code, and we will add type hints in all major classes and functions. An example of these modifications is provided here:
>
> ```python
> @registry.register_dataset("pfdeltadata")
> class PFDeltaDataset(InMemoryDataset):
>    """
>    PFDeltaDataset is a base class for loading and handling PFDelta datasets for
>    power system machine learning tasks. It provides functionality to process
>    and store graph-based data representations of power networks.
>    """
>    def __init__(
>        self,
>        root_dir: str = "data",
>        case_name: str = "",
>        split: str = "train",
>        model: str = "",
>        task: float = 1.3,
>        add_bus_type: bool = False,
>        transform: Callable = None,
>        pre_transform: Callable = None,
>        pre_filter: Callable = None,
>        force_reload: bool = False,
>    ):
> ```
>
>
> > I do not understand the directory structure described in the README. Where can I find the .tar.gz archives? Do I have to generate them first, or can I download them from Hugging Face? Please make this clearer in the README.
>
> Thank you for raising this point. We realized that we had omitted the link to the HuggingFace repository (where the dataset is hosted) from the README in our code repository. The data for each case is publicly available on the HuggingFace link provided in the abstract of our paper. Users can access the data by navigating to the appropriate casename folder and extracting the corresponding .tar.gz file within each topological_perturbation sub-folder. This data can either be downloaded manually or by cloning the HuggingFace repository and unzipping the tar.gz files with standard Unix commands (e.g. tar -xvzf FILENAME.tar.gz). After navigating to the desired subfolder, the compressed files can then be unzipped. We will be sure to update the README in the repository to include more detailed instructions on how to do this. In parallel, we are also working on modifying the PFDeltaDataset class to automatically download and unpack the relevant data for each casename for the final code release.
>
> > Why is the FashionMNIST data set included? (see core/datasets/fashionmnist.py)
>
> Apologies – this was a mistake on our part. FashionMNIST and a simple CNN were used to debug and test the structure of the codebase and training scripts, and we (embarrassingly) forgot to remove the FashionMNIST dataset in our final push. We will remove this in the final version of the codebase.
>
> > Would be nice to provide links/references in the README for the models (e.g., CANOS, GNS, PFNet) and other software packages such as PowerModels
>
> Good point - we will add these links.
>
> > Turn Figure 1-3 into vector graphics (also holds for the Figures in the appendix)
>
> Will do!
>
> > Figure A.2: Caption is cut off
>
> Oops, thanks for catching this mistake. We will fix it!

---

> > ### Comment · Reviewer_AtCg · 2025-08-01
> >
> > Thanks for the clarification. This resolves all my concerns.

---

### Official Review · Reviewer_5Jk4 · 2025-07-03

**Rating:** 5
**Confidence:** 3

**Summary:**

The paper proposes a benchmark focusing on power flow problem under real conditions. The proposed PF3 covers around 700k solved power flow instances across 6 grid sizes and with diverse perturbations in load, generation, and network topology and also the close-to-infeasible cases. The paper also proposes metrics for evaluating methods’ deplorability, interpretability, and scalability. It then conducts experiments for benchmarking the existing GNN-based algorithms and traditional solvers. The results show that GNN methods excel at running speed but fail in term of accuracy compared to traditional solvers, showing the challenges for developing future GNN-based methods.

**Additional Feedback:**

1. In section 4.1, it is shown that the dataset is generated from pure random perturbations. There is no alignment with the real-world distributions. To what extent do current perturbations used in the PF3 benchmark reflect the patterns in the real-world power systems?

2. In Table 4, the runtime of the NR method, although slower than GNN-based methods, seems to be still acceptably low. But the accuracy of GNNs outperforms significantly. Are there any scenarios where runtime is a critical bottleneck, and the GNNs are truly needed for this problem setting?

3. In 153-161 lines, the construction of close-to-infeasible samples is based on pushing the power flow Jacobian to singularity. What is the physical interpretation of these samples? Can this generation method cover all close-to-infeasible situations in the real world?

4. The power flow problem can have multiple solutions. How do authors choose the ground truth answer from these multiple solutions?

5. The current results in the experiment section focus on PBL metrics. Have the authors examined what kind of patterns in the samples lead to the failure of the models?

**Dataset Code Accessibility:**

Yes

**Ethical Considerations:**

No, there are no or only very minor ethics concerns

**Limitations Weaknesses:**

1. The benchmark proposed in the paper lacks alignment with the real-world distributions.

2. The motivation for using GNN-based methods in this problem setting is unclear.

3. The experiments focus on the general performance of the methods but lack a detailed analysis of failure case patterns.

**Strengths Contributions:**

1. The benchmark contains diverse perturbations for the power flow, which makes it comprehensive and close to the real-world situation.

2. The benchmark generates challenge close-to-infeasible cases for evaluating the methods’ robustness under extreme conditions.

3. The evaluation tasks cover in-distribution, out-of-distribution, data efficiency, and challenge cases, which are comprehensive and push the standardization of the evaluation in the community.

4. The benchmarking on GNN-based and traditional methods provides a lot of insights and clearly states the future directions for developing advanced methods.

---

> ### Author Rebuttal · Authors · 2025-07-30
>
> Thank you for your support and thoughtful review.  We’d like to address the following limitations raised by the reviewer:
>
> > The benchmark proposed in the paper lacks alignment with the real-world distributions.
>
> > In section 4.1, it is shown that the dataset is generated from pure random perturbations. There is no alignment with the real-world distributions.
>
> We appreciate the reviewer’s observation and agree that datasets incorporating real-world distributions are highly valuable to the field, particularly for tasks such as time-series forecasting and multi-period OPF. However, the goal of our dataset is to benchmark how well machine learning (ML) models can solve the power flow problem across a wide range of operating conditions.
>
> Specifically, one of our goals is to foster the development of ML methods that are robust across different operational scenarios, given that power flow is used not just in normal operations, but also for applications such as contingency analysis that explicitly test stressed or edge-case conditions. Here, incorporating realistic operational data is not essential and could even be counterproductive if done without care. Real-world data often reflects repetitive historical patterns with strong correlations, which can cause models to overfit rather than being robust across scenarios. Even when abnormal event data is present, it is typically sparse and hence insufficient for learning such behaviors.
>
> As such, our dataset instead focuses on capturing a broader region of the feasible space than captured in normal operations data. By moving beyond the narrow band that historical data usually resides in, we aim to support the development of models that can solve the power flow problem under arbitrary load and generation scenarios. This aligns with the behavior of traditional industry-standard solvers like Newton-Raphson, which can handle arbitrary inputs.
>
> > The motivation for using GNN-based methods in this problem setting is unclear.
>
> The reviewer makes a good point that GNNs are not the only possible ML-based approach for solving power flow. We use GNNs as a starting point because they are the most popular ML approach identified for power flow; they are a natural fit given that power flow is a node value prediction problem and that traditional solvers need to generalize across varying grid sizes (which standard feedforward networks struggle to do). However, other methods can also be a good fit for the problem (e.g. transformers), and our benchmark is in fact readily compatible with other types of methods. Node and edge data are stored as tensors, making it simple to access these attributes and feed them into any standard feedforward network. We will update our submission to note that our benchmark is inclusive to any ML approach for power flow.
>
> > The experiments focus on the general performance of the methods but lack a detailed analysis of failure case patterns.
>
> > The current results in the experiment section focus on PBL metrics. Have the authors examined what kind of patterns in the samples lead to the failure of the models?
>
> Thank you very much for this feedback. We now have done a more detailed analysis of the error patterns demonstrated in Task 1.3, and will include two new figures alongside this analysis in the paper.
>
> The first figure presents histograms of the maximum power balance loss (PBL Max) across all test samples, disaggregated by model (CANOS, PFNet, GNS-S). We focus on standard feasible cases, rather than close-to-infeasible cases, for this initial analysis. Each model exhibits a right-skewed distribution with a long tail, indicating that while the majority of samples have relatively low PBL Max values, a small subset experiences significantly larger violations. The spread and shape of each distribution also differs across models. CANOS has the tightest distribution with the lowest mean and standard deviation, suggesting that it is more consistently accurate across scenarios. PFNet shows a wider spread and slightly higher mean, while GNS-S not only has the widest spread but also exhibits a secondary spike in the distribution, suggesting a bimodal behavior. This bimodal trend in GNS-S indicates that for a subset of samples, the model fails in a qualitatively different way, possibly entering a regime where it generalizes poorly (e.g., due to overfitting to certain perturbation types or failing to capture key constraints). This may reflect instability or sensitivity to certain topologies or loading conditions and warrants further investigation.
>
> The second figure shows histograms of power balance loss (PBL) per node, categorized by bus type (PQ, PV, Slack) for each model. Across all models, PQ buses consistently have lower PBL than PV buses. Some possible interpretations of this behavior include:  (1) reactive power (predicted at PV buses) is harder to predict correctly than voltage magnitude (predicted at PQ buses), or (2) errors in reactive power contribute more heavily to overall PBL than errors in voltage magnitude. Performance on the slack bus varies distinctly across models. CANOS achieves exactly zero PBL on the slack bus by design, due to its analytical treatment of that node. It also has the lowest PV and PQ PBL values in comparison to the other models. In contrast, PFNet and GNS-S display broad distributions of slack bus PBL, indicating greater variability and potential model instability at the slack node.
>
> > In Table 4, the runtime of the NR method, although slower than GNN-based methods, seems to be still acceptably low. Are there any scenarios where runtime is a critical bottleneck, and the GNNs are truly needed for this problem setting?
>
> There are multiple applications where solving power flow becomes a computational bottleneck and obstacle. For example, we are collaborating with a major power grid operator that would like to solve security-constrained optimal power flow (SCOPF) on operational timescales (e.g. in 5 minutes–1 hour), where solving thousands of power flows during the contingency analysis step is the primary computational bottleneck. (Contingency analysis requires solving the power flow problem thousands of times while considering multiple possible outage scenarios, and these analyses need to be run routinely in short periods of time to support critical decision making.) Since grid operators such as these have assessed that NR-based AC power flow is too slow for these workflows, they have resorted to linear approximations such as DC power flow that are much faster but introduce feasibility issues that are increasingly punishing with the integration of low-inertia renewables. Consequently, our grid operator collaborators are looking for cheaper AC power flow solutions to improve realism.
>
> A second major example is topology optimization, which explores discrete actions such as bus splitting and line switching in order to alleviate power grid congestion. This is an increasingly important problem – e.g., one of our grid operator collaborators is facing up to 20 transmission constraints per day. Unfortunately, the topology optimization problem is extremely large – realistic grids may have more than 10^{100} feasible options, with just a single substation on the 118 bus system having e.g. 65,000 possible configurations [1]. Grid operators have begun to explore a variety of methods for this problem – e.g. online optimization and reinforcement learning methods – but underneath, each “search step” relies on running a power flow.
>
> Overall, our collaborations with power grid operators on these and other problems validates the need for faster AC power flow solutions, given that NR-based AC power flow is too expensive to run at the scale needed.
>
> [1] Chauhan, A, et al. PowRL: A reinforcement learning framework for robust management of power networks. AAAI 2023.
>
> > In 153-161 lines, the construction of close-to-infeasible samples is based on pushing the power flow Jacobian to singularity. What is the physical interpretation of these samples? Can this generation method cover all close-to-infeasible situations in the real world?
>
> Close-to-infeasible samples, as constructed in our work, correspond to scenarios where the grid is at its loadability limit. At this point, a further increase in load cannot be accommodated by the grid.
>
> To clarify, our goal is not to capture all possible real-world sources of infeasibility, which also include dynamic phenomena or topology changes. Instead, we propose a method to deliberately generate a specific class of difficult, near-infeasible operating points. Regardless of the source of the infeasibility, close-to-infeasible samples are often underrepresented in typical datasets.
>
> We will explain these points further in the paper.
>
> > The power flow problem can have multiple solutions. How do authors choose the ground truth answer from these multiple solutions?
>
> This is a good point; the existence of multiple solutions motivates our choice of unsupervised power balance loss (PBL) as a main evaluation metric. PBL evaluates the extent to which a model output satisfies the power flow equations, rather than measuring distance from ground truth samples. This contrasts with previous work on OPF/power flow where supervised metrics are used to (incorrectly) judge model outputs based on the distance to a particular feasible solution, rather than calculating whether the problem has been solved.
>
> Nonetheless, to enable supervised methods on this benchmark, we do indeed store a power flow solution per instance as well; this follows standard practice in the PF/OPF literature of storing a single feasible solution [2, 3]. This single solution is the one found by IPOPT when solving the modified OPF problem used in our data generation framework.
>
> [2] Trager Joswig-Jones, et al. OPF-Learn: A framework for AC OPF datasets. IEEE ISGT, 2022
>
> [3] Sean Lovett et al. OPFData: Large-scale AC OPF datasets with topological perturbations. 2024

---

> > ### Comment · Reviewer_5Jk4 · 2025-08-05
> > **Response to the reply**
> >
> > Thank you for the detailed response!
> >
> >  Overall, the reply has well addressed my concerns with the following further questions:
> >
> >  (1) It will further strengthen the credibility of the running time discussion if the paper can include a running time comparison between GNNs and NR methods under different sizes of the power grid. The current bus size is generally quite small.
> >
> >  (2) Since there will be multiple solutions, in reality, how should we select one from them? It might be better to include a discussion on this point.

---

> > > ### Author Response · Authors · 2025-08-06
> > >
> > > Thank you for your support!
> > >
> > > > (1) It will further strengthen the credibility of the running time discussion if the paper can include a running time comparison between GNNs and NR methods under different sizes of the power grid. The current bus size is generally quite small.
> > >
> > > Thank you for the feedback! We will include a runtime comparison between ML models and NR under larger bus sizes than are currently shown in the paper to strengthen the credibility of the runtime discussion.
> > >
> > > > (2) Since there will be multiple solutions, in reality, how should we select one from them? It might be better to include a discussion on this point.
> > >
> > > Thank you for this valuable suggestion! We agree that including a discussion on this point would help clarify how we are selecting a power flow solution when, in theory, multiple solutions exist. We will revise the manuscript accordingly, highlighting the following points:
> > >
> > > - Power flow can indeed have multiple solutions, and a reasonable practice for selecting one is to choose a solution that represents a physically meaningful and operationally relevant point.
> > >
> > > - In practice, solvers such as IPOPT do not attempt to enumerate all possible solutions, as this would require solving the problem multiple times with different initializations, which would incur a large computational overhead to produce a single sample. We instead select the sample the solver converges to as per standard practice in the optimal power flow and power flow literature.
> > >
> > > - Although we do not explicitly recover the full set of power flow solutions, we have verified that our data generation procedure captures a diverse range of operating conditions. As illustrated by the violin plots in the appendix, the dataset includes both high-voltage, small-angle-difference solutions (typical of normal operations) and lower-voltage, larger-angle-difference solutions (which may be useful for studying system stress or instability).

---

### Official Review · Reviewer_L6nC · 2025-07-03

**Rating:** 4
**Confidence:** 4

**Summary:**

This paper introduces PF∆, a benchmark dataset for evaluating machine learning approaches to power flow calculations. The dataset captures diverse variations in load, generation, and grid topology, including near-infeasible cases, and provides standardized tasks and metrics.

**Dataset Code Accessibility:**

Yes

**Dataset Code Comments:**

The dataset is fully accessible and the accompanying code is publicly available.

**Ethical Comments:**

The proposed benchmark dataset focuses solely on technical power system simulations and does not involve human subjects, sensitive data, or potential misuse scenarios that could raise ethical concerns. All experiments are conducted on synthetic grid models with open-source implementations, ensuring transparency and avoiding any societal risks.

**Ethical Considerations:**

No, there are no or only very minor ethics concerns

**Limitations Weaknesses:**

1. The paper "A Large Synthetic Dataset for Machine Learning Applications in Power Transmission Grids" proposed a large-scale synthetic dataset for ML applications in power transmission networks, with methods to adjust the dataset close to N or N-1 feasibility boundaries. What are the innovations of this paper compared to that one?
2. The data generation process relies on existing solvers (e.g., Newton-Raphson). Could the biases of these solvers affect the dataset generation?
3. Should the computational cost of dataset generation be taken into consideration?
4. The dataset primarily focuses on specific grid models (e.g., IEEE systems). Can it be extended to more grid configurations?

**Strengths Contributions:**

1. The paper proposes a standardized set of tasks and metrics for evaluating machine learning models, addressing a key limitation in the current literature.
2. The introduction of PF∆ provides a rich and diverse dataset that captures a wide range of operational conditions, including load variations, generator profiles, and topological perturbations.
3. The dataset includes challenging cases near the limits of voltage stability, which are often difficult for traditional solvers.
4. The paper includes a thorough evaluation of traditional solvers and state-of-the-art machine learning models, providing valuable insights into the strengths and weaknesses of different approaches.

---

> ### Author Rebuttal · Authors · 2025-07-30
>
> Thank you to the reviewer for their thoughtful review. We would like to address the following limitations/weaknesses raised:
>
> > “The paper "A Large Synthetic Dataset for Machine Learning Applications in Power Transmission Grids" proposed a large-scale synthetic dataset for ML applications in power transmission networks, with methods to adjust the dataset close to N or N-1 feasibility boundaries. What are the innovations of this paper compared to that one?”
>
> Thank you for bringing up this interesting paper, which provides a relevant but parallel contribution to our work. This paper proposes a disaggregation method for power consumption to generate large-scale synthetic load datasets that reflect real-world power systems. This yields a time series that resembles actual system behavior, as well as a method to  allow users to tune the level of cross-correlation between load time series across different buses. There are several key differences between this paper and our proposed work. First, their dataset relies on solving a DC power flow based optimization to determine power generation dispatch, which does not  guarantee AC feasibility of samples. In contrast, our dataset produces generation setpoints by using a full AC network model, thus ensuring the feasibility of our samples by construction. Second, while their algorithm has the flexibility to tune datasets closer to the N and N-1 feasibility boundary, they do not describe a methodology that deliberately pursues samples that are at the feasibility boundary. Our dataset, on the other hand, targets near-infeasible and ill-conditioned cases using a continuation power flow. Third, their paper is more-so tailored to time-series tasks, such as anomaly detection in power systems. In contrast, our benchmark is specifically focused on the (single time step) power flow problem, a different and foundational task in power system analysis. We will be sure to cite and add discussion of this paper in our related work section.
>
> > “The data generation process relies on existing solvers (e.g., Newton-Raphson). Could the biases of these solvers affect the dataset generation?”
>
> Thank you for raising this important issue. Solver biases can indeed influence dataset generation – notably, the assessment of which samples are feasible or not relies on the strength of the solver used to make this feasibility assessment. We have taken several steps to mitigate the effects of any biases. First, we chose a robust state-of-the-art solver, IPOPT, to generate our data. This helps reduce the chance that samples are erroneously labeled as infeasible due to solver limitations (and as a consequence not included in the dataset). Moreover, through the use of continuation power flow, we deliberately produce ill-conditioned operating points, which can be often missed in standard sampling approaches due to the difficulty existing solvers have with such cases. Lastly, our overall evaluation metric (PBL) is unsupervised, which means we are directly measuring how well the ML model outputs satisfy power flow, rather than introducing evaluation biases in favor of the specific solutions found by IPOPT.  We will revise the manuscript to explicitly acknowledge this point regarding solver-induced biases, and clarify the steps we have taken to reduce them in our dataset.
>
> > “Should the computational cost of dataset generation be taken into consideration?”
>
> Thank you for raising this point. The premise of this work is to replace online/real-time computational burden with offline computational burden (data generation and training costs), in order to enable better real-time decision-making. While the underlying assumption is that offline computational burden is not the primary bottleneck (compared to online computation time), we agree that it is important to document the offline cost (including data generation). We will include details on computation time in the revised submission.
>
> To give additional context: The motivation for developing this dataset stems from the large number of power flow (PF) calculations used in real-time operations. For instance, the problem of security-constrained optimal power flow ideally needs to be solved in operational timescales (e.g., every 5 minutes or hourly), and the primary computational bottleneck to this process is the need to solve hundreds of PF evaluations (as part of contingency analysis) in a fraction of this time frame. More recently, the problem of topology optimization has been explored as a low-cost solution to alleviate power grid congestion, and this can entail searching over $10^{100}$ topology optimization actions on realistic grids (using, e.g., brute force, optimization, or reinforcement learning) – with each step of the search requiring a PF. Put otherwise, given the sheer number of PF computations that need to be solved online, as well as the speed at which they need to be solved, we believe that the offline investment in data generation and training is well-worth the substantial online efficiency gains. Notably, this only works if models are able to appropriately generalize to operational variations without the need for retraining – hence our focus on generating a diverse dataset, particularly in terms of load profiles and operating regimes.
>
> To give details on some general trends we observed during data generation: We found that as we applied our perturbation scheme to larger grid sizes, the time required to find and generate feasible samples increased significantly. For example, generating 105,000 power flow samples (without close-to-infeasible case generation) for a 500-bus system took approximately 3 days using 3 nodes (each with 72 Intel Xeon E5-2670 CPUs), while generating 52,500 samples for a 2000-bus system required 7 days on the same hardware. As a result, we identified the design of scalable data generation techniques as a promising direction for future research.
>
> > “The dataset primarily focuses on specific grid models (e.g., IEEE systems). Can it be extended to more grid configurations?”
>
> Thank you for raising this important issue, which is not completely clear in the paper. Our data generation pipeline is fully generalizable to any network grid that is saved in a data format that can be parsed by PowerModels. This includes network files  that follow the PSS/E format and MATPOWER network files in the .m format. We will revise the manuscript to make this capability clear.

---

> > ### Author Response · Authors · 2025-08-06
> >
> > Thank you again for your review of our paper. Please let us know if you have any remaining questions or clarifications, which we would be happy and eager to address during the reviewer-author discussion period.

---

> > > ### Comment · Area_Chair_wJW1 · 2025-08-08
> > >
> > > Dear Reviewer,
> > >
> > > Please respond to the author's rebuttal.
> > >
> > > Best regards,
> > >
> > > Your AC

---

### Official Review · Reviewer_wt5U · 2025-07-21

**Rating:** 2
**Confidence:** 5

**Summary:**

The submission introduces PF$\Delta$, a dataset designed for benchmarking power flow solvers under diverse operating conditions. The dataset incorporating variations in load, generation, and network topology, including contingency scenarios (N, N-1, N-2) and near-infeasible cases. The authors benchmark both traditional and machine learning-based solvers on this dataset.

**Additional Feedback:**

1. Incorporate time-series data or realistic temporal sampling to better reflect operational conditions.
2. Extend the dataset to include larger networks (10k+ buses) and document scalability.
3. Clearly define and document the treatment of infeasible cases.
4.Provide support or guidance for adapting the dataset to OPF tasks with custom objectives.

**Dataset Code Accessibility:**

Yes

**Ethical Considerations:**

No, there are no or only very minor ethics concerns

**Limitations Weaknesses:**

1. Limited Realism: Lack of Temporal Correlation
The dataset appears to sample load and generation scenarios independently, without considering the strong temporal correlations present in real-world time series data. As a result, the dataset may not adequately capture operational patterns or challenges that arise in practice.

2. Scale is Not State-of-the-Art
While the dataset covers networks up to approximately 2,000 buses, this falls short of the scale seen in state-of-the-art research and real-world grids, which often exceed 10,000 buses. This limits the dataset’s relevance for large-scale power system studies and may reduce its adoption by practitioners working on industrial-scale problems.

3. Unclear Handling of Infeasible Cases
While the paper mentions “close-to-infeasible” scenarios, it does not clearly explain how truly infeasible cases are handled—whether they are discarded, labeled, or included as negative examples. Additionally, the dataset generation process appears to store only a single feasible solution per instance, despite the possibility of multiple solutions in non-convex power flow and OPF problems.

4. No Support for Custom OPF Objectives
The dataset focuses solely on the power flow (feasibility) problem and does not support optimal power flow (OPF) formulations with custom objectives.

5. ML Model Limitation
There have been quite many work on solving OPF/PF problems using different ML models and algorithms, why the GNN is explicitly considered? How should the audience adapt the dataset/model to other power system tasks?

6. Benchmarking Scope is Limited
Benchmarks are restricted to the provided dataset and do not evaluate scalability to larger or more complex grid models. It is also unclear whether the codebase can be easily extended to accommodate new objectives, network topologies, or data types.

**Strengths Contributions:**

1. Large, Diverse Dataset: Nearly 700,000 power flow solutions across six grid sizes, including challenging and near-infeasible cases.
2. Relevant Benchmarking: Provides comprehensive benchmarks for classical power flow solvers, supporting reproducible research.
3. Community Value: Offers a standardized testbed for evaluating solver performance on realistic scenarios.
4. Clear Motivation: Addresses the need for more robust and varied power flow datasets.

---

> ### Author Rebuttal · Authors · 2025-07-30
>
> Thank you for your detailed review. Before addressing specific concerns, we would like to clarify the problem setting and purpose of this benchmark. This benchmark focuses on power flow, which involves determining all voltages in the system given specified load and generator setpoints. This is related to but distinct from optimal power flow (OPF), which solves for optimal generator setpoints given loads. While the reviewer raises several points related to OPF in their review (e.g., temporal correlations for multi-period OPF, custom OPF objectives), our benchmark is expressly focused on power flow, a challenging and independently valuable single-period simulation task. Power flow calculations are the backbone for real-time applications such as contingency analysis (where repeated power flow calculations assess the viability of an operating point under outages) and for topology optimization (which entails power flow-based search over potentially $10^{100}$ scenarios on realistic grids).
>
> For these applications, solvers must meet several key criteria: (a) fast solution times, (b) satisfaction of power flow equations across diverse and near-infeasible operating conditions (c) generalization to topology changes and new grid sizes (d) robustness to varying amounts of training data (e) scalability to realistic grid sizes, and (f) handling of infeasible cases. These are each independently challenging problems for which current machine learning (ML) methods fall short. Our benchmark does not aim to capture every variation of these challenges, but instead provides an appropriately difficult basis to help bring ML methods to the next level of maturity. We focus primarily on challenges (a)--(d) in this benchmark, as well as somewhat on (e) via support for medium-sized grids, with the lens that maturing existing ML methods along these axes is a crucial first step towards addressing the broader set of challenges.
>
> We would now like to address specific concerns raised by the reviewer:
>
> > “Load/generation scenarios are sampled independently, without considering real-world temporal correlations and operational patterns.”
>
> We appreciate the reviewer’s observation and agree that real-world datasets with temporal correlations are highly valuable, e.g. for time-series forecasting and multi-period OPF. However, the goal of our dataset is to benchmark how well ML models can solve the (single time step) power flow problem.
>
> Our goal is to foster ML methods that are robust to diverse operational scenarios and for applications such as contingency analysis that explicitly test edge-case conditions. Here, incorporating realistic operational time series data is not essential and could even be counterproductive if done without care. Real-world time series data often reflects repetitive historical patterns with strong correlations, which can cause models to overfit to this “common case” data rather than being robust across scenarios. Even when abnormal event data is present, it is typically sparse and insufficient for learning such behaviors.
>
> As such, our dataset instead captures a broader region of the feasible space than seen in normal operations data. By moving beyond the narrow band that historical data usually resides in, we aim to support the development of models that can solve power flow under load and generation diversity. This aligns with the behavior of industry-standard solvers like Newton-Raphson, which handle arbitrary inputs, regardless of whether they reflect any temporal correlations.
>
> > “With networks capped at ~2,000 buses, the dataset falls short of the >10,000-bus systems seen in real-world grids/cutting-edge research.”
>
> We acknowledge that good performance on very large grids is important; however, there are two reasons why we did not exceed grid sizes greater than 2000 buses. First, we found that even state-of-the-art models like CANOS were not performing well on smaller grids, such as 118 bus systems. Thus, we designed our benchmark to “meet the methods where they are,” to foster initial advancement towards the ultimate goal of meriting testing on large systems. Second, data generation is time consuming, particularly when providing sufficient generator and load diversity – e.g., creating 52,500 feasible power flow samples for a 2000-bus system took 7 days on 3 nodes (each with 72 Intel Xeon E5-2670 CPUs) without including close-to-infeasible case generation. This is likely why most large-scale benchmark datasets for the closely related optimal power flow (OPF) problem are for grids of ~100 buses, or, if they have larger network sizes, the data generation process does not capture sufficient diversity in feature values. PFΔ is the first dataset to simultaneously introduce perturbations across generation, loads, and topologies to generate data. To our knowledge, a diverse data generation scheme that scales to realistic grid sizes is yet to be found, making it a promising direction for future research.
>
> > “The paper references “close-to-infeasible” scenarios but does not clarify how truly infeasible cases are treated.”
>
> In our framework, we discard infeasible samples as stated in line 127. However, we acknowledge that for many applications, being able to categorize whether a sample is infeasible or not is an important co-requisite to performing well on feasible cases. As ML models still struggle on the latter, this benchmark focuses on advancing ML predictions on feasible cases (including on close-to-infeasible cases which is novel compared to previous work), leaving the problem of handling truly infeasible cases to future/parallel work.
>
> > “The dataset stores one feasible solution per instance, despite potential for multiple solutions in non-convex power flow and OPF problems.”
>
> This is a valid point and motivates our use of the power balance loss (PBL) as an unsupervised metric; our model evaluation is based solely on the extent to which the power flow equations are satisfied, rather than distance from ground truth samples. We felt this was important precisely because multiple solutions can exist in power flow, and that supervised metrics used in some previous work on OPF/power flow are actually (incorrectly) judging distance to a particular feasible solution rather than whether the problem has been solved. However, to provide support for training supervised methods on this benchmark, we store a single supervised solution per instance as well; this follows standard practice in the PF/OPF literature of finding a single feasible solution, rather than exhaustively identifying all possible ones [1, 2].
>
> [1] Trager Joswig-Jones, et al. OPF-Learn: A framework for AC OPF datasets. IEEE ISGT, 2022.
>
> [2] Sean Lovett et al. OPFData: Large-scale AC OPF datasets with topological perturbations, 2024.
>
> > “The dataset focuses solely on the power flow (feasibility) problem and does not support optimal power flow (OPF) formulations with custom objectives.”
>
> We agree that developing benchmark datasets for OPF is a valuable research direction. However, our dataset is designed with a different purpose: to support progress on the power flow problem, which (unlike OPF) is not an optimization task, but a simulation task given inputs of some (potentially non-optimal) conditions. Power flow is an independently important problem to address for its applications in (e.g.) scenario analysis, contingency analysis, and topology optimization. With that said, we believe that the study of ML for power flow can also benefit work on ML for OPF (e.g., challenges such as robustness to different loading scenarios, generalization, and data efficiency are shared).
>
> > “With many ML approaches for OPF/PF, why focus specifically on GNNs?”
>
> The reviewer makes a good point that GNNs are not the only possible ML-based approach for addressing power flow. We use GNNs as a starting point because they are the most popular ML approach identified for power flow; they are a natural fit given that power flow is a node value prediction problem and that traditional solvers need to generalize across varying grid sizes (which standard feedforward networks struggle to do). However, other methods can also be a good fit for the problem (e.g., transformers), and our benchmark is readily compatible with other types of methods. Node and edge data are stored as tensors, making it simple to access these attributes and feed them into any standard feedforward network. We will update our submission to note that our benchmark is inclusive to any ML approach for power flow.
>
> > “How can users adapt the dataset/model to other power system tasks?”
>
> As described above, our focus on power flow is intentional due to its independent value and because it embodies ML challenges relevant to other power system tasks. For instance, generalization over topologies has been relatively underexplored in the OPF literature (due to emphasis on solution feasibility), but we believe that solutions and insights developed for power flow on PF∆ can yield direct insights for OPF and related tasks as well.
>
> > “Benchmark scope is confined to the provided dataset, with unclear adaptability to new objectives, grid sizes, topologies, or data types.”
>
> Thank you for raising this important issue, which is not completely clear in the paper. First, the data generation pipeline in our codebase is fully generalizable to any power network data that can be parsed by PowerModels. This includes network files in PSS/E format and MATPOWER network files in the .m format. We will revise the manuscript to make this clear. Second, our codebase can accommodate user-designed custom loss functions for training and different network topologies/data types in the input data. In addition (while not intended as a main contribution), data generation functions in the codebase can be adapted for e.g. OPF with custom objectives (such as transmission loss minimization).

---

> > ### Author Response · Authors · 2025-08-06
> >
> > Thank you again for your review of our paper. Please let us know if you have any remaining questions or clarifications, which we would be happy and eager to address during the reviewer-author discussion period.

---

> ### Comment · Area_Chair_wJW1 · 2025-08-08
>
> Dear Reviewer,
>
> Please respond to the author's rebuttal.
>
> Best regards,
>
> Your AC

---

### Note · Authors · 2025-08-14

We want to thank the reviewers and the AC for their time and consideration. As part of our final remarks, we emphasize the importance of this benchmark and highlight three key points relevant to our rebuttal discussions. The power flow (PF) problem determines voltages and power generation in the system given specified load and generator setpoints, a challenging and valuable single-period simulation task. PF calculations are the backbone of important real-world applications, such as contingency analysis and topology optimization, that require solving hundreds to thousands of PF instances under strict time constraints or outside of normal operating conditions. Machine learning tools could address this bottleneck, but must meet key criteria: (a) fast solution times, (b) accuracy across diverse and near-infeasible conditions, (c) generalization to topology changes and new grid sizes, (d) robustness to limited training data, (e) scalability to realistic grids, and (f) handling of infeasible cases. Our benchmark spans challenges (a)--(d), and partially (e), to provide a difficult testbed to advance ML for PF methods toward greater maturity.
* Diverse Scenarios: By representing a broader region of the feasible space than seen in normal operations data, our dataset fosters ML methods that are robust to diverse scenarios, and aligns with industry standard solvers like Newton-Raphson. We intentionally do not consider real-world data, as it often reflects repetitive historical patterns, which can cause models to overfit to this “common case” data.
* Flexible Benchmark/Dataset: Our data generation pipeline supports any power network data parsable by PowerModels, e.g. PSS/E and MATPOWER .m files. If needed, data generation functions in our codebase can be adapted for OPF with custom objectives, e.g., transmission loss minimization. We focus initial testing on medium-sized grids to match the current maturity of ML methods, with the flexibility to extend to large systems once scalable methods for generating diverse data are developed.
* Unsupervised Evaluation: To mitigate solver bias when multiple solutions exist, our data generation uses the solver only (1) to determine whether a solution is feasible, and (2) to save said solution to enable supervised methods. Using an unsupervised metric allows us to evaluate models based on how well outputs satisfy the PF equations, rather than their distance from a solver’s solution.

Additional points are covered in the rebuttals.

---

### Decision · Program_Chairs · 2025-09-18

**Decision:**

Accept (poster)

**Comment:**

This paper introduces PFΔ, a large-scale benchmark for machine learning approaches to the power flow (PF) problem, with ~700k diverse instances spanning load, generation, and topology variations, including near-infeasible cases. The dataset supports standardized evaluation with unsupervised metrics (PBL) and is benchmarked against both classical solvers and state-of-the-art GNNs. The contribution is timely: PF is a core bottleneck in contingency analysis and topology optimization, and PFΔ provides the first systematic testbed designed to advance ML methods beyond small-scale or narrow-scope datasets.

The reviews were mixed. One reviewer argued for rejection, citing lack of temporal correlations, limited grid size (<10k buses), unclear infeasible handling, and an overly GNN-centric framing. The authors convincingly clarified scope (PF, not OPF), explained infeasible treatment and unsupervised evaluation, documented dataset generation costs, and showed that the benchmark is architecture-agnostic. Other reviewers were positive, praising novelty, comprehensiveness, and community value, and their remaining concerns (runtime scaling, multiple solutions, repo clarity) were acknowledged with clear camera-ready commitments. Overall, while realism and industrial-scale coverage remain future directions, PFΔ is technically sound, well-motivated, and reproducible, with clear community impact. I recommend acceptance.